



# Volcanic ash forecast using ensemble-based data assimilation: the Ensemble Transform Kalman Filter coupled with FALL3D-7.2 model (ETKF-FALL3D, version 1.0)

Soledad Osores[1,2,3], Juan Ruiz[4], Arnau Folch[5], and Estela Collini[6]

[1]Servicio Meteorológico Nacional (SMN), Buenos Aires, Argentina
[2]Consejo Nacional de Investigaciones Científicas y Técnicas (CONICET), Buenos Aires, Argentina
[3]Comisión Nacional de Actividades Espaciales (CONAE), Buenos Aires, Argentina
[4] Centro de Investigaciones del Mar y la Atmósfera, Facultad de Ciencias Exactas y Naturales, Universidad de Buenos Aires, CONICET, UBA. UMI-IFAECI (CNRS-CONICET-UBA). Departamento de Ciencias de la Atmòsfera y los Océanos, Facultad de Ciencias Exactas y Naturales, Universidad de Buenos Aires. Buenos Aires, Argentina , Argentina
[5] Barcelona Supercomputing Center (BSC), Barcelona, Spain
[6] Servicio de Hidrografía Naval (SHN), Buenos Aires, Argentina

**Correspondence:** Soledad Osores (msosores@smn.gob.ar)

**Abstract.**

Quantitative volcanic ash cloud forecasts are prone to uncertainties coming from the source term quantification (e.g. eruption strength or vertical distribution of the emitted particles), with consequent implications on operational ash impact assessment. We present an ensemble-based data assimilation and forecast system for volcanic ash dispersal and deposition aimed at reducing

5   uncertainties related to eruption source parameters. The FALL3D atmospheric dispersal model is coupled with the Ensemble Transform Kalman Filter (ETKF) data assimilation technique by combining ash mass loading observations with ash dispersal simulations in order to obtain a better joint estimation of 3D ash concentration and source parameters. The ETKF-FALL3D data assimilation system is evaluated performing Observation System Simulation Experiments (OSSE) in which synthetic observations of fine ash mass loadings are assimilated. The evaluation of the ETKF-FALL3D system considering reference

10   states of steady and time-varying eruption source parameters shows that the assimilation process gives both better estimations of ash concentration and time-dependent optimized values of eruption source parameters. The joint estimation of concentrations and source parameters leads to a better analysis and forecast of the 3D ash concentrations. Results show the potential of the methodology to improve volcanic ash cloud forecasts in operational contexts.

## 1   Introduction

15   Volcanic ash dispersal forecasts are routinely used to prevent aircraft encounters with volcanic ash clouds and to define flight re-routed trajectories avoiding potentially contaminated airspace areas. In the aftermath of the 2010 Eyjafjallajökull volcano eruption in Iceland, safety-based quantitative criteria for air traffic disruption were introduced, originally based on ash concentration thresholds and, more recently, on engine ingested dose/dosage (Clarkson et al., 2016). These scenarios involve the implementation of quantitative ash concentration forecasts, which require of better model input constrains, particularly on ash





emission rates and/or on model initialization. A large amount of scientific research has been conducted in recent years to: i) better quantify the amount of ash emitted, its vertical distribution across the column and the related uncertainties; ii) obtain data on the 3D structure of ash clouds, particularly using ground, aircraft, and space-based instrumentation; iii) improve model representation of physical processes occurring within ash plumes and clouds and; iv) transfer of scientific outcomes into op-

erations. However, despite the substantial advances in model formulation and initialization, it is estimated that, in operational contexts, forecasted ash concentrations still can have an uncertainty as large as one order of magnitude (e.g., IVATF, 2011).

Epistemic uncertainties in ash dispersal forecasts may have different origins, including: i) uncertainties in the source term (i.e. eruption column height, mass eruption rate, particle grain size distribution); ii) uncertainties in the atmospheric model driving dispersal simulations (e.g. wind velocity and direction, small scale turbulence intensity, atmospheric temperature and

humidity) and; iii) uncertainties in model parameterizations of physical processes occurring both in the eruptive column and during subsequent passive transport (e.g. ash settling and removal processes, particle aggregation, interaction with meteorological clouds, etc.). In addition to these, aleatoric uncertainties exist always regarding the future evolution of the Eruption Source Parameters (ESPs) when an eruption is on-going at the time of running a forecast. Several studies (e.g., Zehner, 2010; Kristiansen et al., 2012) have concluded that the main source of epistemic uncertainty in ash dispersal forecasts comes from

the ESPs that, very often, are not well constrained in real time.

Inverse modeling and, in particular, data assimilation methods, are techniques that can be used to estimate the state of dynamical systems based on partial and noisy observations. In a broad sense, these techniques build on continuous or quasi-continuous observations to produce model initial conditions (analyses) that can be used to better predict the future state taking into account uncertainties in observations and model formulation. Data assimilation methods have been successfully applied to

the estimation of the state of the ocean or the atmosphere (e.g., Kalnay, 2003; Carrassi et al., 2018) as well as for the optimization of uncertain model parameters (e.g., Ruiz et al., 2013). More recently, applications have been extended to atmospheric constituents (e.g., Bocquet et al., 2010; Hutchinson et al., 2017), including ash dispersion models with the purpose of estimating the 3D distribution of ash concentrations to be used as initial conditions for forecasts. Surprisingly, examples of application of data assimilation techniques to volcanic ash dispersion are scarce and still mainly limited to a research level. For example,

Wilkins et al. (2015) implemented a data insertion methodology to improve the initial conditions of ash concentrations based on satellite estimations of ash mass loadings in a Lagrangian dispersion model. Fu et al. (2015, 2017a) applied an Ensemble Kalman Filter technique to the estimation of ash concentrations in an Eulerian dispersion model based on flight concentration measurements and satellite estimations using idealized experiments and real observations. Their results showed that both observational sets (flight measurements and satellite mass loads) reduced forecasts errors, in their particular case attributed to a

wrong model representation of ash sedimentation processes. One important issue when using satellite estimates of ash mass loadings is that observations only provide a 2-D distribution of ash mass while models usually require the vertical profile of ash concentrations. Fu et al. (2017b) presented a modified approach for the comparison between model and observations in the context of the ensemble Kalman filter that try to deal with this limitation.

Uncertainties in the source parameters can be circumvented in part by using inverse modeling techniques for the optimization

of these parameters. Eckhardt et al. (2008) implemented a source parameter optimization approach based on the definition of





a cost function which measures the departure of ash concentrations from observed values and the departure of the estimated parameters from their a-priori values. This allowed to reconstruct the full emission profile using data from different sensors. Stohl et al. (2011); Kristiansen et al. (2012); Denlinger et al. (2012); Pelley et al. (2015); Steensen et al. (2017) discussed further developments and evaluations of the proposed approach. In particular, Pelley et al. (2015) describe the operational

implementation of this algorithm at the London VAAC. In Chai et al. (2017), the optimal parameters are found using a quasi-Newton minimization approach of a similar cost function and Lu et al. (2016) use a similar approach in the context of an Eulerian model. Finally, Zidikheri et al. (2017a, b) presented an optimization algorithm based on a systematic search of the optimal parameter values for both qualitative and quantitative ash forecasts and evaluate the performance of the technique for different cases showing a positive impact on forecast quality. Wang et al. (2017) performs idealized experiments in which

a particle filter and an expectation maximization algorithm are used for the estimation of ash source parameters obtaining promising results.

The goal of this paper is to contribute on the development of data assimilation methods to improve quantitative ash dispersion forecasts. To this end, we propose an ensemble-based data assimilation system for volcanic ash combining an Ensemble Transform Kalman Filter (ETKF) (Ott et al., 2004; Hunt et al., 2007) and the FALL3D ash dispersal model (Costa et al., 2006;

Folch et al., 2009), named ETKF-FALL3D. This system produces a joint estimation of 3-D ash concentration and critical ESPs, that can improve the performance of the classical ash dispersion forecast strategies. This manuscript presents a first analysis of the ETKF-FALL3D system using different Observing System Simulation Experiments (OSSEs) in which synthetic observations of ash column mass loadings are simulated and assimilated. The system is evaluated under constant and time-dependent ESPs and the sensitivity of the system performance to parameter uncertainty, ensemble size, and observations uncertainty is

explored and discussed. Additionally, some impacts of the Gaussian assumptions underlaying the ensemble Kalman filter in the present case are discussed. A description of the methodology is presented in section 2, the experimental setup of the sensitivity experiments is described in section 3, the results are discussed in section 4, and the final conclusions are outlined in section 5.

## 2   Methodology

### 2.1   The FALL3D model

FALL3D is an Eulerian atmospheric dispersal model that solves the advection-diffusion-sedimentation equation for a set of particle classes (bins), each characterized by a particle size, density and shape factor. Given an eruption source term and meteorological variables, FALL3D solves the 4D ash concentration for each particle class, from which the total and the fine ash column mass loadings are diagnosed performing a vertical integration. The meteorological fields must be furnished off-line by a Numerical Weather Prediction (NWP) model forecast or from a re-analysis dataset. The source term determines the amount

of tephra injected to the atmosphere, its vertical distribution along the eruption column, and the fraction of mass associated to each particle bin. This term can be parameterized using different schemes available in the model for the Mass Eruption Rate (MER) (e.g., Mastin et al., 2009; Degruyter and Bonadonna, 2012; Woodhouse et al., 2013) and the vertical mass distribution (e.g., Pfeiffer et al., 2005; Folch et al., 2016). For simplicity and without loss of generality, we will assume here a MER given





by the Mastin et al. (2009) scheme, which depends on the fourth power of the eruption column height and does not account for wind effects, and a Suzuki vertical mass distribution (Pfeiffer et al., 2005) depending on two shape parameters $A$ and $\lambda$. The Suzuki parameter $A$ controls the relative height at which emission is maximum whereas the parameter $\lambda$ controls the width of the distribution of mass around that level. A previous sensitivity test (Osores, 2018) has shown that the two FALL3D model

parameters that affect most the model results are the eruption column height $h$ and the parameter $A$ in the Suzuki distribution. For this reason, these two parameters will be used in the following sections to define the ETKF-FALL3D system experiments.

## 2.2    The ETKF-FALL3D System

In operational applications, data assimilation is implemented sequentially to provide an estimation of the state of the system at a series of times in the so-called "data assimilation cycle". Each data assimilation cycle consists on two steps: a first step in

which the numerical model is used to provide an *a priori* estimation or forecast of the state of the system and its uncertainty, followed by a second step in which the prior estimation is combined with observations (which are also considered uncertain), to obtain a posterior estimation or analysis. These two steps are repeated sequentially in order to propagate forward in time information from past observations.

Let us assume that the state of a system at time $t$ is represented by a state vector $\boldsymbol{x_t}$ that, in our particular case, consists on

the values of ash concentration at each model grid point and for each particle class. In other words, $\boldsymbol{x_t}$ is a column vector with $n$ elements, being $n$ the total number of state variables in the FALL3D model (*i.e.* the total number of grid points times the number of particle bins). For parameter estimation, model parameters $\boldsymbol{\theta}$, e.g. those defining the characteristics of the source term, are also considered part of the state of the system and thus assumed uncertain. For the sake of simplicity, we limit the FALL3D source term parameters to the eruption column height $h$ and the Suzuki distribution $A - Suzuki$ parameter, but the

methodology that follows can easily be extended to any other set of model input parameters. The augmented state vector $\boldsymbol{s_t}$ at time $t$ is defined as the concatenation of the state vector $\boldsymbol{x_t}$ and the (time-dependent) estimated model parameters $\boldsymbol{\theta}$, that is, $\boldsymbol{s_t} = [\boldsymbol{x_t}, \boldsymbol{\theta_t}]$ is a column vector with $n_s = n + 2$ elements.

In the Ensemble Kalman filter the time-dependent uncertainty in the state variables and parameters is estimated using a Monte-Carlo approach through an ensemble of augmented states. Let us assume that we start at time *t-1* with an ensemble of

initial conditions and model parameters. Then, our forecast of the state of the system at time *t*, is obtained by integrating in time the FALL3D model for each ensemble member:

$$\boldsymbol{s_t^{f(i)}} = M_t(\boldsymbol{x_{t-1}^{a(i)}}, \boldsymbol{\theta_{t-1}^{a(i)}}), \tag{1}$$

where $M_t$ represents the FALL3D model operator which integrates the model in time for the *i-th* ensemble member starting from the *i-th* initial conditions (analysis) $\boldsymbol{x_{t-1}^a}$ and fixing the model parameters to $\boldsymbol{\theta_{t-1}^a}$ during the time integration interval.

Note that a persistence model is assumed for the model parameters (*i.e.* $\boldsymbol{\theta_t^f} = \boldsymbol{\theta_{t-1}^a}$) since no information about its variations is available yet during the forecast. Following the assumptions of the ensemble Kalman filter, the joint *a priori* probability





distribution of the augmented state at time $t$ is assumed Gaussian, with a mean and a covariance matrix estimated from the ensemble of forecasts:

$$\overline{s_t^f} = k^{-1} \sum_{i=1}^{k} s_t^{f(i)} \tag{2}$$

$$\mathbf{P_t^f} = (k-1)^{-1} \mathbf{S_t^f S_t^{f^T}} \tag{3}$$

where $\overline{s_t^f}$ is the ensemble forecast mean, $\mathbf{P_t^f}$ is the ensemble forecast covariance matrix (a square matrix of dimension $n_s \times n_s$), and $\mathbf{S_t^f}$ is the ensemble forecast perturbation matrix whose $i$-th column is computed as $\mathbf{S_t^{f(i)}} = \mathbf{s_t^{f(i)}} - \overline{\mathbf{s_t^{f(i)}}}$.

Note that the integration of the ensemble in time propagates the uncertainty on the initial conditions and parameters at time *t-1* into the future in order to provide a time dependent estimation of the forecast uncertainty. This is a key feature that makes these methods particularly appealing for the estimation of uncertain model parameters (e.g. Aksoy et al., 2006; Ruiz et al., 2013) and for an accurate quantification of concentration.

At the analysis step a set of observations is available which is related to the true state of the system by the following expression:

$$\boldsymbol{y}_t = \mathcal{H}(\boldsymbol{x}_t^{true}) + \boldsymbol{\epsilon}_t \tag{4}$$

where $\boldsymbol{y}_t$ is a *m*-size column vector containing the value of the *m* observations at time *t* and $\boldsymbol{x}^{true}$ is the true model state (assumed to be unknown). $\mathcal{H}$ is a (usually non-linear) transformation that maps the state variables (i.e. ash concentrations for different particle sizes) into the observed quantities (e.g. cloud column mass load) and the vector $\boldsymbol{\epsilon}$ represents the error in the observations. This error is typically assumed to be a zero-mean Gaussian random variable with covariance matrix $\mathbf{R}$ (dimensions of *m x m*). The errors in the observations are assumed to be uncorrelated in time and independent of the state of the system. Under these assumptions, the information provided by the forecast and the observations can be combined to obtain an estimation of the augmented state that minimizes the root mean square error with respect to the unknown truth (e.g., Kalnay, 2003; Carrassi et al., 2018):

$$\overline{s_t^a} = \overline{s_t^f} + \mathbf{P_t^f H_t^T} (\mathbf{H_t P_t^f H_t^T} + \mathbf{R})^{-1} (\boldsymbol{y}_t^f - \mathcal{H}(\boldsymbol{x}_t^f)) \tag{5}$$

where $\overline{s_t^a}$ is the a posteriori estimation of the augmented state (*i.e.* the analysis) and $\mathbf{H_t}$ is the tangent linear of the observation operator. The factor $\mathbf{P_t^f H_t^T} (\mathbf{H_t P_t^f H_t^T} + \mathbf{R})^{-1}$ is usually referred as the Kalman gain. The Kalman Filter equations also provide a way to estimate the uncertainty of the analysis. After the assimilation of the observations, the augmented state covariance matrix is updated to:

$$\mathbf{P_t^a} = (\mathbf{I} - \mathbf{KH_t}) \mathbf{P_t^f} \tag{6}$$


where $\mathbf{P_t^a}$ is the posterior or analysis augmented state covariance matrix. Note that (5) and (6) can be used to obtain an ensemble of analyses for the state variables and the parameters whose ensemble mean is equal to $\overline{s_t^f}$ and the perturbations are sampled from a Gaussian distribution with zero mean and covariance matrix equal to $\mathbf{P_t^a}$. These equations can be difficult to solve explicitly for high dimensional systems due to the large size of $\mathbf{P_t}$ and $\mathbf{R_t}$ but several methods have been proposed to

address this issue and to implement the ensemble Kalman Filter in high dimensional systems. In the present work, we use the ETKF approach which solves the ensemble Kalman Filter equations in a sub-space defined by the ensemble members. Details about the equation that arises from this particular implementation can be found in Hunt et al. (2007), but a summary is given in Apendix A. One of the main advantages of this approach is that finding the analysis ensemble mean requires to invert a $k \times k$ matrix, which is significantly cheaper than inverting the $n \times n$ matrix for the case in which $k \ll n$ (which is usually the case for

high dimensional applications of the filter).

The process is schematically shown in Figure 1. The cycle starts with an estimation of the mean parameters, assuming they have a Gaussian distribution, $k$ random samples are taken. Each parameter sample is used in one of the ensemble members integrated with the dispersion model. When an observation is available, it is combined with the ensemble forecasts using the ETKF equations. From this combination an ensemble of analysis is obtained with a set of optimized parameters that also has a

Gaussian distribution. Then the next cycle starts from the ensemble of analysis and the set of optimized parameters, to produce a new ensemble forecast. When a new observation is available, the assimilation method is applied, and the cycle continues so on.

## 3   ETKF-FALL3D experimental set-up

To explore the capability of the ETKF-FALL3D system we use an OSSE approach, in which a long model integration is

performed and regarded as the true evolution of the ash cloud. This model integration will be referred as the nature run. Observations are simulated from the nature run and then assimilated with the ETKF-FALL3D system. The June 2011 Puyehue-Cordón Caulle eruption (Osores et al., 2012; Collini et al., 2013) has been selected for the generation of the nature run.

### 3.1   Ash mass loading observation simulations

The nature run and observation simulation begins at 18:00 UTC 4[th] June 2011 and lasts for 10 days up to 00:00 UTC 15[th] June

covering the domain shown in Fig. 2 with a model horizontal resolution of 0.23° and a vertical resolution of 200 meters. The model top is located at 20 km above the ground. The volcanic vent is located at 40.52ºS - 72.15ºO at an altitude of 1420 m a.s.l.

The Particle Total Grain size distribution (GSD) is represented by 12 classes with diameters between 2 mm (-1$\phi$) and 1$\mu$m (10$\phi$) and densities ranging between 400 for the larger particles to 2100 $\mathrm{kg\ m^{-3}}$ for the smaller ones (Bonadonna et al., 2015). The vertical distribution of the source is parameterized using the Suzuki scheme considering $\lambda$=5, the settling velocity model

is that of Ganser (Ganser, 1993), and the vertical and horizontal turbulent diffusion are parametrized by the similarity (Ulke, 2000) and CMAQ (Byun and Schere, 2006) schemes respectively. The meteorological fields are obtained from the Global



Forecasting System (GFS) analysis with an horizontal resolution of 0.5º, a temporal resolution of 6 hours and 27 constant pressure vertical levels.

The simulated observations represent ash mass column loads (vertically integrated ash mass per unit area) estimates retrieved from satellite radiances (e.g., Prata and Prata, 2012; Francis et al., 2012; Pavolonis et al., 2013). Simulation of satellite retrievals

are chosen since these observations are available almost globally and have a high spatial and temporal resolution making them particularly appealing for the implementation of operational data assimilation systems. To represent some of the limitations of current satellite-based ash mass load retrievals, the simulated observations are available only where the true load values are between $0.2\,\mathrm{g\,m^{-2}}$ and $10\,\mathrm{g\,m^{-2}}$ (e.g., Wen and Rose, 1994; Prata and Prata, 2012; Pavolonis et al., 2013). The observational error is assumed to have a random Gaussian distribution, with a standard deviation of 0.15 of the ash mass load.

For the sake of simplicity observations are assumed to be co-located with the model grid points, we also assume that observation errors are uncorrelated (i.e. $\mathbf{R}$ is diagonal) and that observations are unbiased. All observations are generated assuming a clear sky condition both above and below the ash cloud. Two nature runs were generated to evaluate the ETKF-FALL3D system: one with constant emission profiles and another with time-varying emission profiles.

### 3.1.1 Constant emission profile

This nature run simulation considers a source term that remains constant during all the simulated period, with an eruption column height of 8.5 km above the vent and a $A - Suzuki$ parameter of $5.5$ (Figure 3a). Figure 3b shows the ash mass loading from the nature run on 7$^{th}$ June at 12:00 UTC for illustrative purposes. The number of available observations (which depends on the thresholds described in the previous section) is time dependent (ranging from 27 to 52 grid point observations) and, in this particular case, is primarily affected by the atmospheric circulation which produces variations in the 3D ash concentration

within the model domain.

### 3.1.2 Variable emission profile

In this experiment, $h$ and $A - Suzuki$ are time dependent (Fig. 3a). In order to represent a realistic variability of the source parameters, the $h$ evolution corresponds to the estimated heights during the 2011 Puyehue-Cordón Caulle eruption (Osores et al., 2014). Since the $A - Suzuki$ parameter can not be directly estimated, the evolution of this parameter is simulated using

an autoregressive model (Fig. 3).

In Fig. 3c, the 7$^{th}$ June at 12:00 UTC ash mass loading field is shown. In this experiment the number of observations that are assimilated depends on the emission profile as well as the wind field, and it can range from 15 (on 11$^{th}$ June 06:00 UTC) to 86 (on 11$^{th}$ June at 18:00 UTC).

### 3.2 Data assimilation experiments setup

In the data assimilation experiments performed in this work, the simulated observations are assimilated every 6 hours. The number of ensemble members in the experiments is set to 32 (unless stated otherwise). In most experiments source parameters





are assumed to be unknown and estimated within the data assimilation cycle. The model grid, boundary conditions and all other model parameters and configuration options are as in the nature run. The ensemble at the first assimilation cycle is initialized using zero ash concentrations for all members and a set of parameters which are sampled randomly from a Gaussian distribution. The relaxation to prior spread inflation approach (RTPS, Whitaker and Hamill (2012)) with a parameter of $\alpha = 0.5$

is applied to the state variables to reduce the impact of sampling error. For the parameters the conditional inflation approach of Aksoy et al. (2006) is implemented which prevents the parameter ensemble spread to fall below a prescribed threshold. Since the domain used in the data assimilation experiments is small, no covariance localization is used in the estimation of the state variables or parameters.

Given that in the ensemble Kalman filter the distribution of ash concentration and parameters is assumed to be Gaussian,

negative ash concentration or unphysical parameter values can result from the assimilation of observations. These unphysical solutions must be corrected before using the analysis ensemble as initial conditions for the next ensemble forecast cycle. For ash concentration, negative values are turned into zero concentrations. In the case of eruption source parameters, unphysical values are replaced by a random sample from the analysis ensemble mean and covariance. If the randomly sampled value is still outside the physically meaningful range for the parameter, the process is repeated. The physically meaningful range for

model parameters is set to 0-20 km and 0-15 for $h$ and $A - Suzuki$ respectively.

One of the main hypothesis of the Kalman filter is that state variables and parameters are approximately linearly correlated with the observations. This is not true for the $h$ parameter since in the Mastin et al. (2009) emission scheme the source strength is proportional to the fourth power of $h$. For this reason, instead of estimating $h$, we estimate $h^4$ so that the estimated parameter is more linearly correlated with the observations.

In this work, several experiments are performed to study the convergence of the filter and its sensitivity to some key parameters. Two experiments are performed using the constant parameter nature run to assess filter convergence. The first experiment starts with source parameters that are higher than the true value and will be referred as CONSTANT-UPPER, the second starts with an under estimation of the source parameters and will be referred as CONSTANT-LOWER. The initial parameter spread for $h$ and $A - Suzuki$ are 500 m and 0.5 respectively and is the same for both experiments. These experiments are com-

pared against an experiment in which parameters remain constant at their initial value (CONSTANT-NOEST) and against an experiment in which the parameters are constant and their ensemble mean is equal to the true value (CONSTANT-TRUE).

A second set of experiments are based on the nature run with time-dependent parameters. An estimation experiment which uses the same parameter ensemble spread as in the previous experiments is performed and will be referred as the CONTROL experiment. To evaluate the impact of performing parameter estimation in the time dependent parameter context, an experiment

in which the parameters are kept constant at the time average of the true parameters is also presented (CONTROL-NOEST).

To quantify the sensitivity of the ETKF-FALL3D system to the parameter ensemble spread, two additional experiments are performed: one in which the ensemble spread is larger than in the CONTROL experiment (HI-SPREAD) where the spread in $h$ and $A - Suzuki$ is 2000 m and 4 respectively, and another experiment in which the ensemble spread is lower than in the CONTROL run (LOW-SPREAD) where the spread in $h$ and $A - Suzuki$ is 100 m and 0.1 respectively. All the other

configuration settings are as in the CONTROL experiment.





To explore the impact of modifying the ensemble size an experiments with ensemble sizes of 8 (ENS-8) and 16 (ENS-16) are presented where all other configuration settings are equal to the CONTROL run experiment. Finally, the impact of observation error is assessed in 2 experiments with observation errors which are of 30 (OBS-30) and 40% (OBS-40) of the true total mass concentration. All presented data assimilation and parameter estimation experiments are summarized in Table 2. Finally, a set
of simulation experiments are carried out using a larger domain to evaluate the impact of the optimized parameters upon the simulation of the ash cloud farther from the vent.

### 3.3 Performance metrics

The evaluation of the FALL3D-ETKF system is achieved by comparing the 3D ash concentration forecast (and analysis) against the nature run, and also by measuring the consistency between the estimated and the actual forecast uncertainties. The
comparison is based on the RMSE, error bias and the ensemble spread of either the forecast or the analysis which are given by the following expressions:

$$RMSE = \sqrt{N^{-1} \sum_{i=1}^{N} (\overline{x}_{f,i} - x_{t,i})^2} \tag{7}$$

$$BIAS = N^{-1} \sum_{i=1}^{N} (\overline{x}_{f,i} - x_{t,i}) \tag{8}$$

$$SPREAD = \sqrt{N^{-1} \sum_{i=1}^{N} (k^{-1} \sum_{j=1}^{k} (x_{f,i}^{(j)} - \overline{x}_{f,i})^2)} \tag{9}$$

where $\overline{x}_{f,i}$ is either the forecast or analysis ensemble mean ash concentration at time and location $i$ and $x_{f,i}^{(j)}$ and $x_{t,i}$ are their corresponding values for the $j - th$ ensemble member and the nature run respectively. Spatial or temporal averages are obtain by summing over $i$.

## 4 Results

### 4.1 Constant emission profile experiments

In these experiments we explore the impact of data assimilation and parameter estimation in the steady parameter scenario. Fig. 4 shows the ensemble mean and the spread of $h$ and $A - Suzuki$. After the first assimilation cycle both parameters start to converge rapidly to values close to the true ones, with mean errors below $500$ m and $1$ respectively. Convergence of $h$ is faster, likely due to the strongest sensitivity of forecasted ash concentrations to column height in the surroundings of the source. The two experiments considering different initial parameter values (CONSTANT-UPPER and CONSTANT-LOWER) converge to





values close to the true parameter, indicating that the parameter estimation technique is robust in finding the correct values of parameters regardless of ensemble initialization. As observed in Fig. 4, both parameter estimation experiments tend to sub-estimate the values of $h$ and to slightly overestimate the values of $A - Suzuki$. Fig. 4 shows also the parameter ensemble spread. In these experiments, the ensemble almost always contains the true parameter value, meaning that the parameter

uncertainty is well captured by the ensemble. However, it should be noted that, in these experiments, the ensemble spread of the model parameters is prescribed *a priori* to a value which may not be the optimal one under different conditions (e.g. if the optimal parameters are time dependent or if other sources of uncertainty like errors in the atmospheric circulation, are present ). Sensitivity experiments to the parameter ensemble spread will be discussed in the following sections.

Figure 5a shows the time evolution of the domain-averaged RMSE for the 3-D total ash concentration forecasts. The RMSE

of the parameter estimation experiments is compared against an experiment in which parameters are not estimated and are fixed at the initial value of the CONSTANT-UPPER experiment (CONSTANT-NOEST) and against an experiment in which the parameter ensemble is centered at the true value of the source parameters (CONSTANT-TRUE). Parameter estimation experiments show similar results in terms of the 6-hour forecast errors, indicating the robustness of the convergence to the optimal parameter values. Moreover, both parameter estimation experiments show ash concentration errors that are similar to the one

obtained in the CONSTANT-TRUE experiment and are much lower than the errors obtained in the CONSTANT-NOEST experiment, clearly showing the advantage of performing data assimilation based source parameter estimation. Figure 5b shows the spatially averaged ash concentration ensemble spread. As observed, the spread values are close to the RMSE values in Figure 5a, which indicates that after convergence of the parameters, the ensemble spread is closely representing the magnitude of the errors.

Figure 5c shows the horizontal and time averaged error bias for the total ash concentration as a function of height. The first two days have been excluded because are considered as part of the spin-up time of the filter. This figure shows that biases associated with the estimation experiments are much lower than for the CONSTANT-NOEST experiment, showing once again the advantage of optimizing the source parameters. The CONSTANT-UPPER, CONSTANT-LOWER and CONSTANT-TRUE experiments show a small systematic underestimation of the maximum concentrations and an over estimation above and below

the location of the maximum. Note that the bias is slightly lower in the parameter estimation experiments with respect to the CONSTANT-TRUE experiment.

The fact that a biased parameter ensemble (i.e. the underestimation of $h$ observed in Fig. 4a) produces a less biased estimation of ash concentrations (Fig. 5c) may be related to the non-linear relationship between $h$ and the total mass emission at the source. Since the emitted mass depends on $h^4$, positive perturbations in $h$ are associated with a much larger emission rate and are thus

farther from the observations than ensemble members with negative perturbations in $h$. This creates a bias in the estimation of the concentrations because, even if the ensemble is centered at the true $h$ value, positive perturbations are farther from observations than the negative ones and therefore the ensemble mean tends to over-estimate concentrations. ETKF tries to compensate for this effect converging to a slightly biased parameter set which reduces the error bias and the RMSE.

As observed in Figure 5d, the analysis error in ash concentration is below the forecast error. This indicates that the ETKF

method is efficient in reducing the error in the 3-D concentration field based in the information provided by a 2-D observation.





This is a remarkable result in a context where most of the observations are 2-D whereas operational requirements are 3-D. This finding will be the base to use the analysis as a better diagnose of the state of the plume to improve the forecasts. The reason behind this lies in the structure of the forecast error covariance matrix, which is estimated from the ensemble of forecasts. This matrix contains the information about the covariances between mass loading (which is the observable quantity) and the

concentration at different heights from which the mass loading is obtained and which are not directly observed. In this work, reliable covariances between 3D ash concentrations and mass loadings are obtained by taking into account the uncertainties associated with the source parameters.

## 4.2 Time dependent emission experiments

These experiments use the observations simulated from the nature run with time-varying parameters (Fig. 3). The parameter

ensemble is initialized with a mean $h$ of 11 km and a mean $A - Suzuki$ of 7 and standard deviations of 0.5 km and 2.0 respectively. Figure 6 shows the evolution in time of the optimized parameter ensemble as well as their corresponding true values, showing a good agreement. The estimation of $h$ seems to be particularly accurate and is able to detect rapid variations in the eruptive column height having RMSE values lower than 200 meters throughout the experiment. The $A - Suzuki$ parameter is also well estimated although the evolution in time of the true parameter is not reproduced so accurately. There are also two

sudden jumps in the estimation of $A - Suzuki$, indicating a less well constrained parameter value. These differences in the behavior of the estimated $h$ and $A - Suzuki$ may be due to the higher sensitivity of the ash distribution to the eruptive column height in comparison with the $A - Suzuki$ parameter. The jumps in the estimated $A - Suzuki$ occur during periods of fast changes in $h$, suggesting that when $h$ is not well-estimated, $A - Suzuki$ may be modified in an attempt to compensate for errors in $h$.

Fig. 7 shows the RMSE of the forecast for the 3D total ash concentration. Errors in this case vary strongly with time, with larger errors corresponding to the instants in which $h$ is larger, leading to stronger ash mass emission at the vent and consequently larger ash concentrations in the surroundings of the vent. The ensemble spread (Fig. 7b), although smaller than the error (indicating an under-dispersive ensemble), changes accordingly with more spread during the periods in which the emission is higher. These changes in the ensemble spread are a consequence of the relationship between $h$ and mass emission

at the vent. Since $h$ deviations from the ensemble mean are almost time independent, the associated departures in mass emission are larger during the periods of higher $h$, leading to a larger spread in the concentration field.

Fig. 7d shows the spatially averaged reduction in the RMSE for the total ash concentration between the forecast and the analysis. The RMSE is reduced between the forecast and the analysis at almost all vertical levels, indicating that the vertical covariance structure between mass loadings and ash concentrations at different levels is well estimated leading to accurate 3D

ash concentration estimations.

In order to assess the impact of treating the parameters as a time dependent variable, this experiment is compared with an experiment in which data assimilation is performed but only the ash concentration field is updated. In this case, source parameters are keep constant in time at a value which is equal to the time average of the true parameters (CONTROL-NOEST, Fig. 6). This value is chosen in order to obtain a solution that is as close as possible to the one obtained with the time dependent parameters.





Fig. 7 shows that the forecast RMSE and bias in the 3D ash concentration is almost always larger in the CONTROL-NOEST experiment with respect to the CONTROL experiment. The error in the CONTROL and CONTROL-NOEST experiments becomes similar around day 3 and after day 8 because at those time instants the source parameters are close to each other (Fig. 6). Moreover, the ensemble spread for the CONTROL-NOEST experiment is almost constant in time and, because of

that, changes in the forecast uncertainty are not captured (Fig. 7b). This is because time variations in the ensemble spread are mainly associated with changes in the mean values of parameters. These experiments suggest that performing data assimilation for the estimation of 3D ash concentrations is not sufficient to properly constrain 3D ash concentration values and that source parameters have also to be taken into account, particularly close to the source where these parameters rapidly impact on concentrations.

As an example, Fig. 8 shows the ensemble forecast mean for the CONTROL and CONTROL-NOEST experiments and for the nature run at FL200 at the 12$^{th}$ assimilation cycle. The ash concentration pattern at this particular level is well represented by the simulation that estimates the source parameters whereas, in the CONTROL-NOEST experiment, there is a significant underestimation of the concentrations due to the underestimation of the column height at this particular time. Note that data assimilation is being performed to correct the 3-D ash concentrations in both experiments.

## 15   4.3   Sensitivity experiments

This section discusses the sensitivity of the analysis and the forecast to the parameter ensemble spread, the ensemble size, and the observation uncertainty. The purpose is to identify which are the potentially more important tuning parameters for the optimization of the system and how robust the system is with respect to errors in observations, which are known to exist in satellite-based ash mass loading estimations.

To explore the sensitivity to the parameter ensemble spread, the experiments CONTROL, HI-SPREAD and LOW-SPREAD with different parameter spreads (Table 2) are compared. Figure 9 shows the estimated $h$ obtained in these experiments as well as the total ash concentration RMSE and bias. As observed, the CONTROL experiment gives the more accurate estimation of $h$ and the minimum RMSE and bias. When the parameter ensemble spread is larger than in the CONTROL experiment, parameter values are systematically under-estimated. As previously discussed, this can be explained by the non-linear dependence

between $h$ and the total emitted mass. However, what is relevant from this experiment is that increasing the ensemble spread degrades the quality of the estimation and increase the impact of non-linearities. Higher dispersion in $h$ increase the magnitude of positive $h$ perturbations leading to a larger error bias, particularly above and below the maximum concentration (Fig. 9c). In the case of the LOW-SPREAD experiment, results are closer to the CONTROL experiment. However, this experiment shows a slower convergence with larger $h$ estimation errors during the first days of the experiment. Slower or lack of convergence

is expected when the parameter uncertainty is under estimated. In this case the ETKF does not allow for large corrections in the parameter values based on the observations, basically because the error in the parameters is assumed to be small. These experiments show that the system is particularly sensitive to the parameter ensemble spread that has to be specified a priori. Moreover, in these idealized experiments, the optimal parameter ensemble spread is determined by the uncertainty in the observations and with no information regarding the changes of the true parameters in time.




As discussed in Section 4.1, parameters are estimated based on their covariance with the observed quantities. In the ensemble based data assimilation methods these covariances are estimated directly from the ensemble, so they can be affected by sampling errors. To asses the impact of these sampling error on the analysis, quality assimilation experiments with different ensemble sizes have been performed. Three experiments with 8, 16 and 32 ensemble members are presented (ENS-8, ENS-16 and
CONTROL respectively). Fig. 10 shows the results in terms of $h$ estimation and total ash concentration RMSE and bias. The CONTROL experiment shows more accurate $h$ estimation and, consistently, lower RMSE and bias values. However, results are not very sensitive to the size of the ensemble. The lack of sensitivity to the ensemble size might be surprising, particularly considering that no spatial localization is being used in order to reduce the impact of sampling errors. However, note that in this case the only source of uncertainty in the system comes from the uncertain parameters. Based on this, the effective dimension
of the space in which uncertainties has to be constrained is two. This is confirmed by the strong covariances that exist between the parameters and ash concentration within the domain (not shown). This effective low dimensionality is reinforced by the fact that the domain is small and close to the source and, because of that, ash concentration at most grid points is strongly correlated with the value of the uncertain source parameters.

The last sensitivity experiment looks into the issue of observation errors in satellite retrievals of mass loadings. In the
experiments presented so far, the standard deviation of the observation errors has been assumed to be 15% of the mass loading in the nature run. However, in real cases, uncertainties associated to mass loading estimations can be larger than that. Two additional experiments are performed to explore the impact of the magnitude of the observation errors on the estimation of source parameters and total ash concentrations with observation standard deviation of 30 (OBS-30) and 40% (OBS-40) of the true mass loading value. Results from these experiments are presented in Fig. 11. As expected, the best results are obtained
with the lowest observation error. However, one interesting result is that as the observation error increases, estimated $h$ values are lower, eventually leading to substantial under-estimations as the ones seen for the OBS-40 experiment during the first days of the experiment. Moreover, these systematic underestimation of $h$ produce an underestimation of the total ash concentrations as visible in the bias profiles (Fig. 11c ). Under the hypothesis of the ensemble Kalman filter, an increase in the observation error leads to an increase in the RMSE of the estimation. However, in this case, the systematic component of the error is also
increased. This behavior is probably a consequence of the non-linear effects arising from the non-linear relationship between $h$ and ash emission rate that has been previously discussed.

### 4.4 Ash concentration simulations in an extended domain simulation

The experiments discussed so far have been performed in a relatively small domain surrounding the vent. In most applications however, it is expected that forecasts over larger domains are required. In this section we explore the adequacy of the parameter
estimation approach to generate a good estimation of ash dispersion over larger domains in an idealized context in which the atmospheric flow is perfectly known. To this purpose a nature simulation over a larger domain (Fig. 12) is performed. This nature run is forced with the same evolution of parameters of the time dependent parameter nature run and spanning the same time period.





To see if the estimated parameters can be used to reconstruct the ash cloud far from the source, the ensemble mean estimated parameters from the CONTROL experiment are used to produce an estimation of the ash cloud. Figure 12 shows a snapshot of the nature run and the experiment forced with the estimated parameters. Note how both ash clouds are very similar even far from the source, indicating that the estimated parameters are sufficient for the reconstruction of the ash plume.

Figure 12 also shows ash distribution at 12 and 24 hours forecast times. These forecasts are initialized from the model run forced with the estimated parameters and use the estimated parameters at the initialization time from the CONTROL run. Parameters remains constant during the forecast since there is no predictive model available for these parameters. Fig. 12 shows that there is a good agreement between forecasts and nature run. This suggests that initializing a forecast from a long run forced with the optimized parameters can be a cost-effective strategy to generate short lead ash concentration forecast over a relatively

large domain.

Although these results are encouraging, it should be taken into account that in more realistic situations, other sources of uncertainty (e.g. uncertainty in the flow or model errors) can significantly affect the evolution of the ash plume far from the source. In this case, the forecast quality can suffer from the estimation of the 3D ash concentration over the entire domain based on the assimilation of mass loading observations.

## 15    5    Summary and conclusions

In this work we have presented an ensemble-based data assimilation system coupled with the FALL3D ash dispersion model that is able to use ash mass loading observations to simultaneously estimate the 3D ash concentration field and to constrain source term parameters. An OSSE preliminary evaluation of the system has been presented and some sensitivity tests performed. The experiments focused on two FALL3D model parameters, one that defines the vertical emission profile and the

eruptive column height (and related emitted mass). The ETKF-FALL3D system shows a robust convergence to the optimal parameters and an accurate reconstruction of the 3-D ash concentrations based on noisy and partial 2-D ash mass loading observations.

Estimation of time-dependent source parameters has been successful within the OSSE context. The ensemble not only produces an estimation of the covariances between the observed variables and the parameters, but also provides a time-dependent

estimation of the forecast uncertainty which resembles the time evolution of the forecast errors. The strong time variability of the ensemble spread is mainly associated with the relationship between column height and emitted mass.

Sensitivity experiments has been conducted to investigate how the parameter ensemble spread, the ensemble size and the observation errors affect the results. The parameter ensemble spread produces a significant impact on the quality of the estimated concentrations and parameters. Larger ensemble spreads lead to stronger biases, both in concentrations and parameters,

whereas lower ensemble spreads produce an over-confident ensemble and slower converge rates that degrade the estimation results. It is important to note that the optimal parameter ensemble spread can depend on the time variability of the estimated parameters and on other sources of uncertainty like errors present in the observations and in the model. The sensitivity to the ensemble size revealed that, even for this low dimensional estimation problem, ensemble sizes up to 32 members show some





improvement with respect to ensembles of 16 and 8 members although the impact of increasing the ensemble size is smaller than the impact associated with changes in the parameter ensemble spread.

The sensitivity to the observation errors shows a particular behavior with an increase in systematic errors both in the parameters and in the concentrations with increasing observational errors. When observation errors reach 40% of the true ash

loadings, the estimated parameters fail to converge during the first days of the experiment leading to significantly larger errors in the ash concentration forecasts.

Experiments presented in this work are limited to a small domain surrounding the vent. Experiments on a larger domain, show that the optimized parameters can be used to force an ash dispersion simulation that can reproduce the ash cloud properties far from the vent as long as the atmospheric circulation is accurately known. This simulations can be used to initialize ash

dispersion forecasts over larger domain as a computationally cheaper alternative to running a data assimilation system with covariance localization over a large domain.

Several research directions are needed from this work, including: a) The improvement of the ETKF-FALL3D system by the application of covariance localization, allowing for a more efficient and accurate estimation of the ash concentrations over larger domains; b) The inclusion of more uncertainty sources in the design of the filter, being the uncertainty in the

atmospheric flow and in the model formulation among the most important; c) The assessment of the skill of the system in more realistic scenarios using real observations; d) A better representation of the uncertainty associated with observations, considering possible covariances among observations as well as systematic biases in the observations; e) The development of techniques that can converge to the optimal parameter ensemble spread based on the information provided by the observations (e.g., Miyoshi, 2011), and (f) The implementation of non-linear assimilation approaches (e.g., Bocquet et al., 2010) that can

better handle non-Gaussian error distributions and non-linear relationships between the model parameters and the observable quantities.

*Code and data availability.* FALL3D model is available through an open license (http://datasim.ov.ingv.it/models/fall3d.html). The ETKF-FALL3D code is written in python. The code and the required data to run a sample experiment are available through an open license at https://doi.org/10.5281/zenodo.3066310. Atmospheric state data from the Global Forecasting System produced by the National Centre for

Environmental Prediction is available through the University Corporation for Atmospheric Research data archive (https://rda.ucar.edu/datasets/ds335.0/, Accessed 07 Apr 2019)





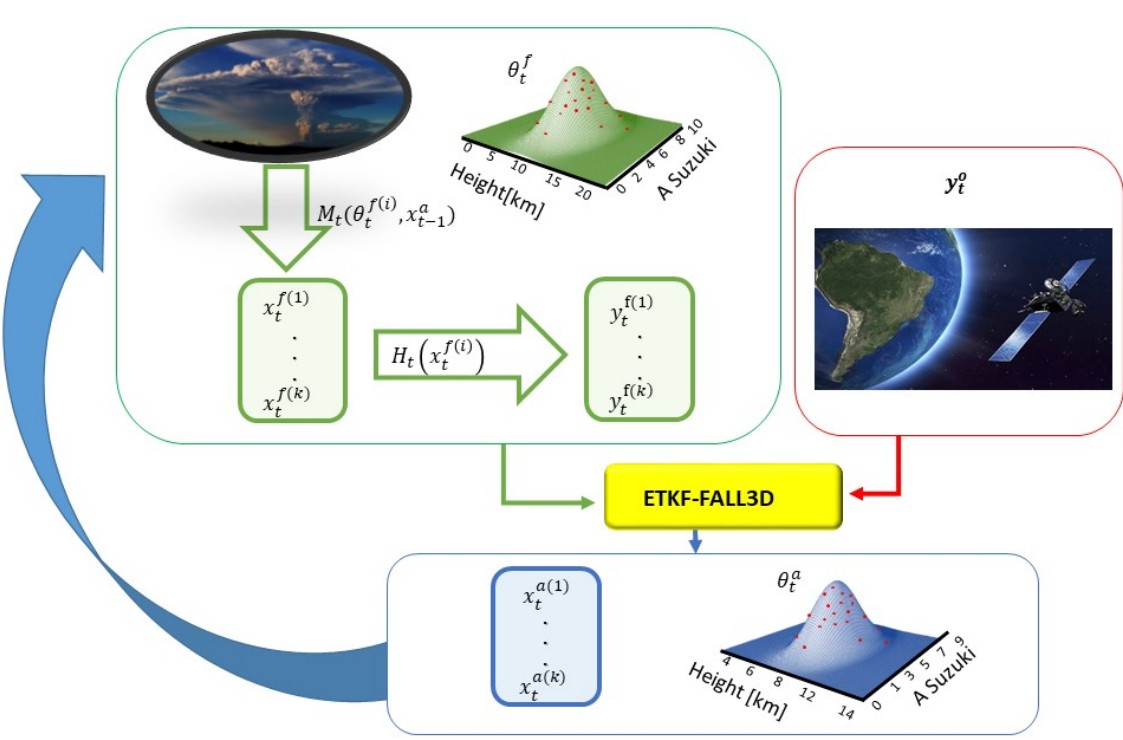

**Figure 1.** ETKF-FALL3D data assimilation system scheme for volcanic ash.



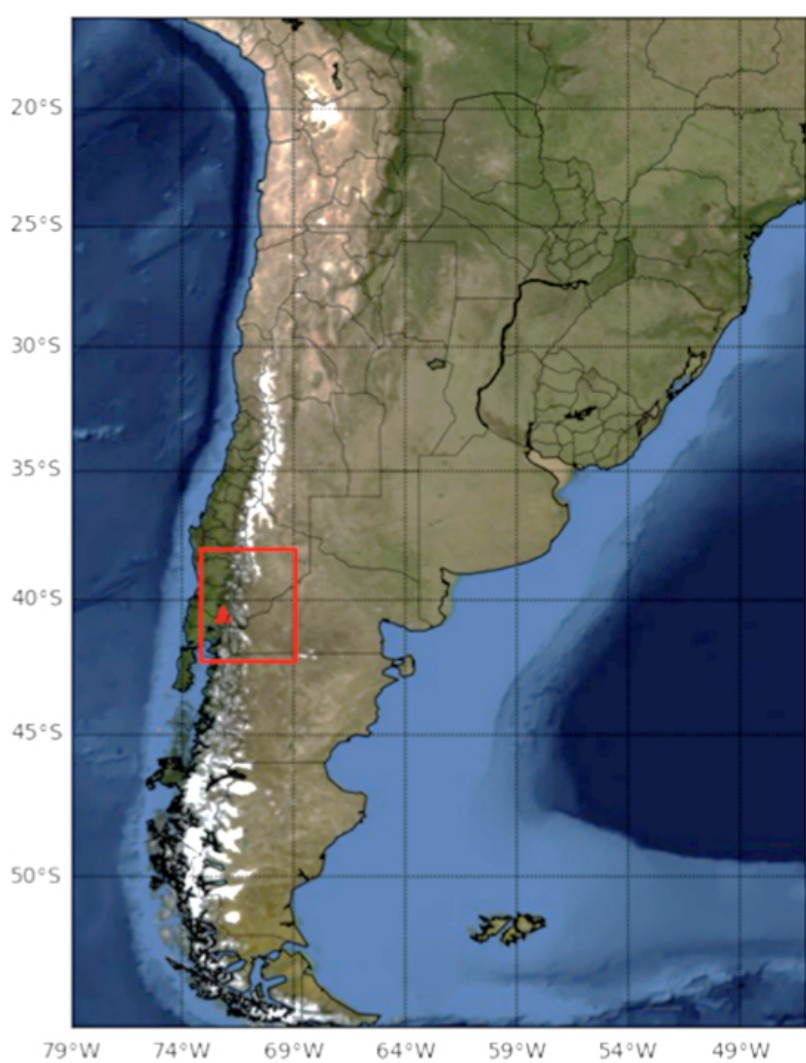

**Figure 2.** Domain used in the ETKF-FALL3D sensitivity tests (red square).



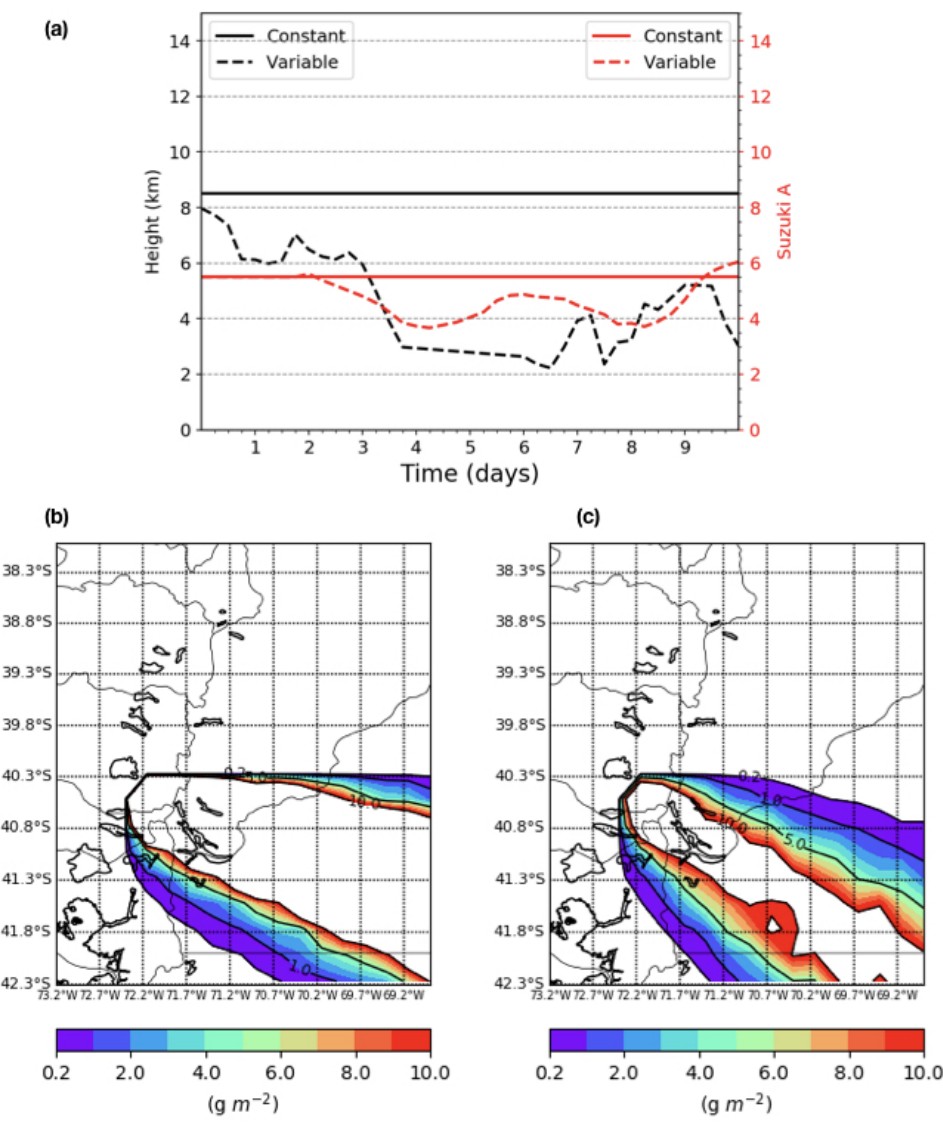

**Figure 3.** (a) Nature run parameters time series for the constant (solid lines) and variable emission profiles (dashed lines) and for $h$ (black lines) and $A - Suzuki$ (red lines). Ash mass loading on 7th June 12:00 UTC with for the (b) constant parameters nature run and the (c) time dependent parameters nature run. Ash mass loading values outside the $0.2 - 10.0 \mathrm{gm}^{-2}$ interval are masked out.





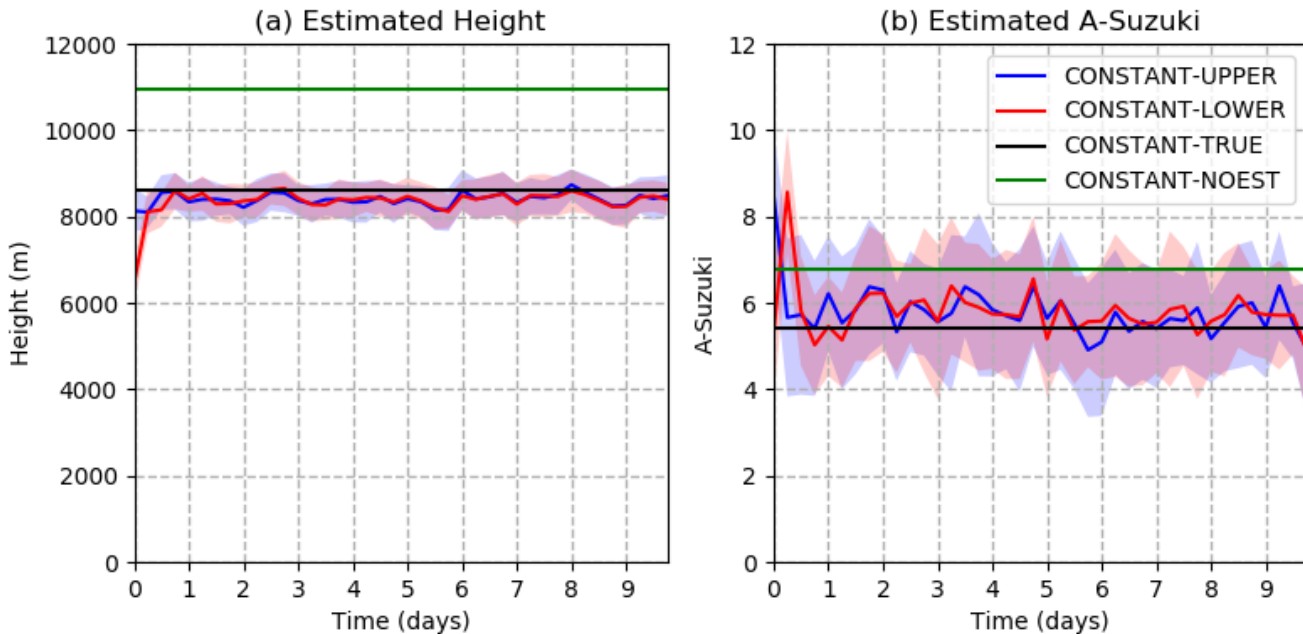

**Figure 4.** Optimized parameters as a function of time in the CONSTANT-UPPER (blue line), CONSTANT-LOWER (red line), CONSTANT-TRUE (black line) and CONSTANT-NOEST (green line) experiments. The shade surrounding the CONSTANT-UPPER and CONSTANT-LOWER estimated values represents +/- one standard deviation. (a) $h$ parameter and (b) $A - Suzuki$ parameter.





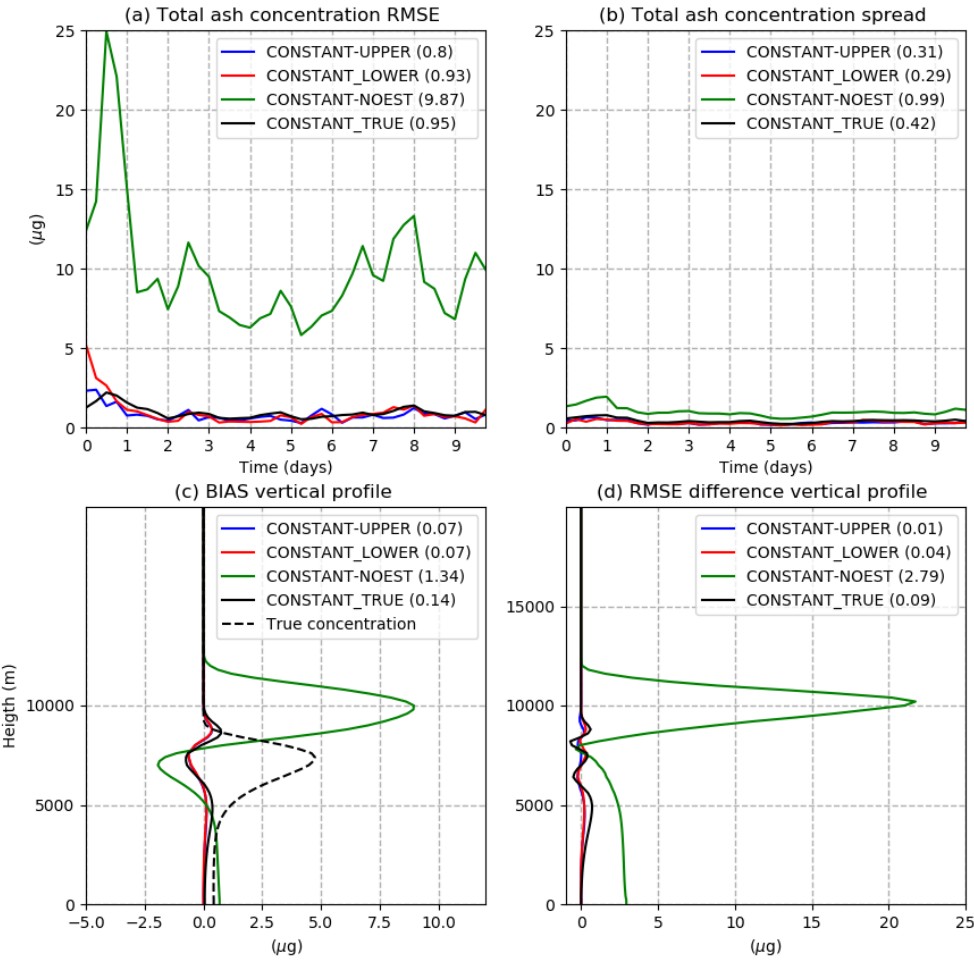

**Figure 5.** (a) Spatially averaged forecasted total ash concentration RMSE, (b) spatially average forecasted total ash concentration ensemble spread, (c) spatially average forecasted total ash concentration bias and (d) difference between the 6-hour forecast and analysis total ash concentration RMSE, for the CONSTANT-UPPER (blue line), CONSTANT-LOWER (red line), CONSTANT-TRUE (black line) and CONSTANT-NOEST (green line) experiments.(a), (b) and (c) are computed from the 6-hour ensemble forecast.





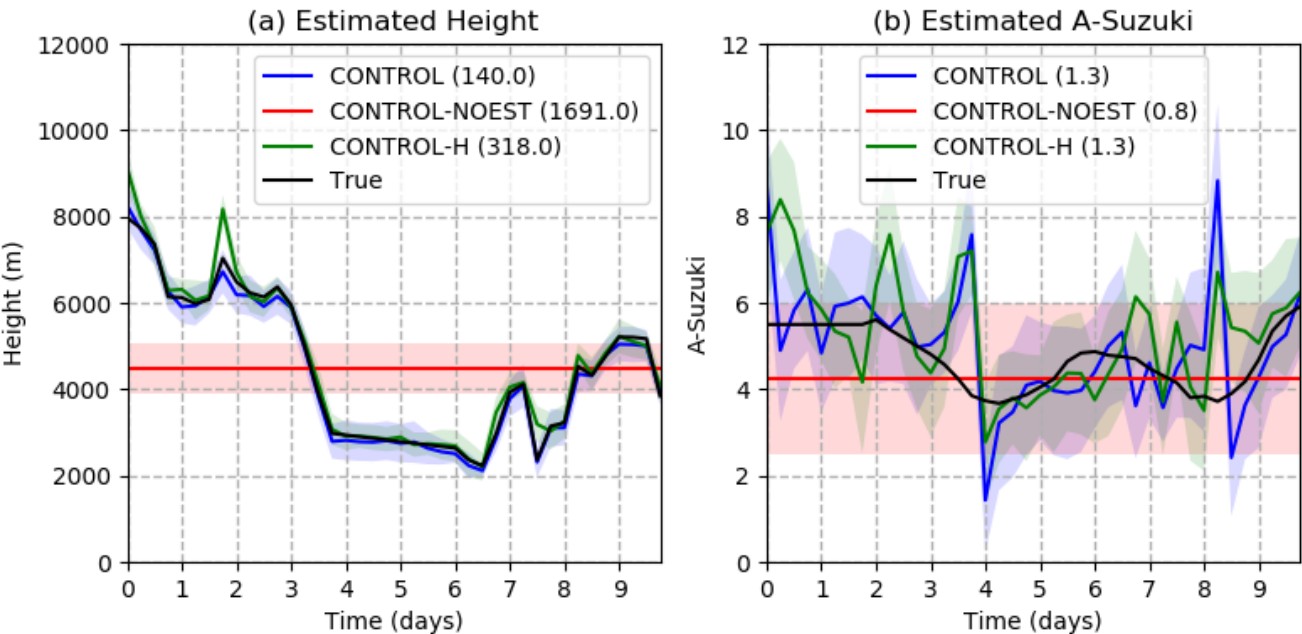

**Figure 6.** Optimized parameters as a function of time in the CONTROL (blue line) and CONTROL-NOEST (red line) experiments. The shade surrounding the estimated values represents +/- one standard deviation. (a) $h$ parameter and (b) $A-Suzuki$ parameter. The black line indicates the value of the parameters in the true run.





**Figure 7.** As in Figure 5 but for the experiments CONTROL (blue line) and CONTROL-NOEST (red line)



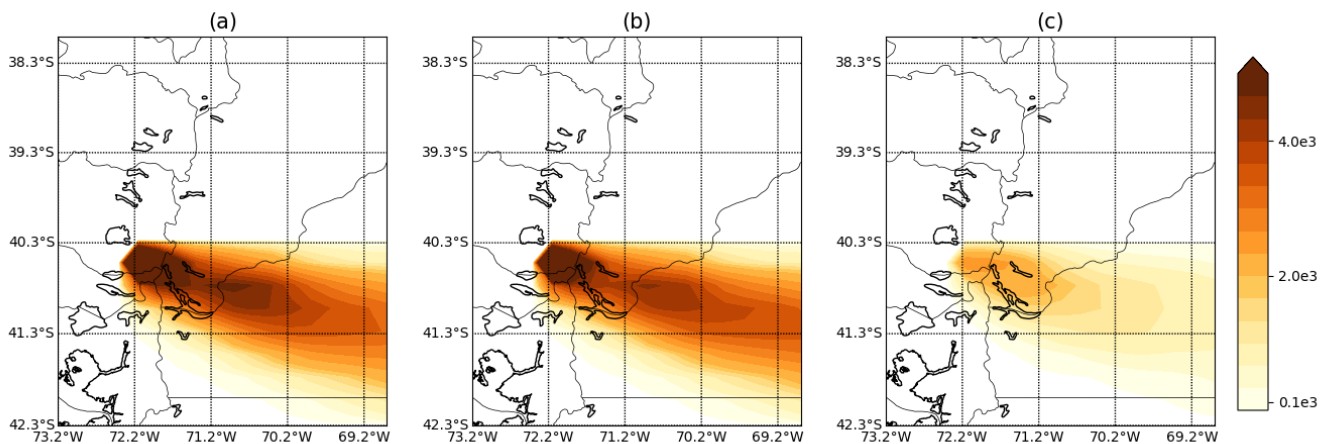

**Figure 8.** (a) Nature run ash concentration at flight level 200 (shaded, $\mu g^{-1}$), (b) as in (a) but for the 6-hour forecast of ash concentration initialized from the CONTROL analysis experiment, (c) as in (b) but for the NOOPT experiment, corresponding to the $12^{th}$ assimilation cycle.



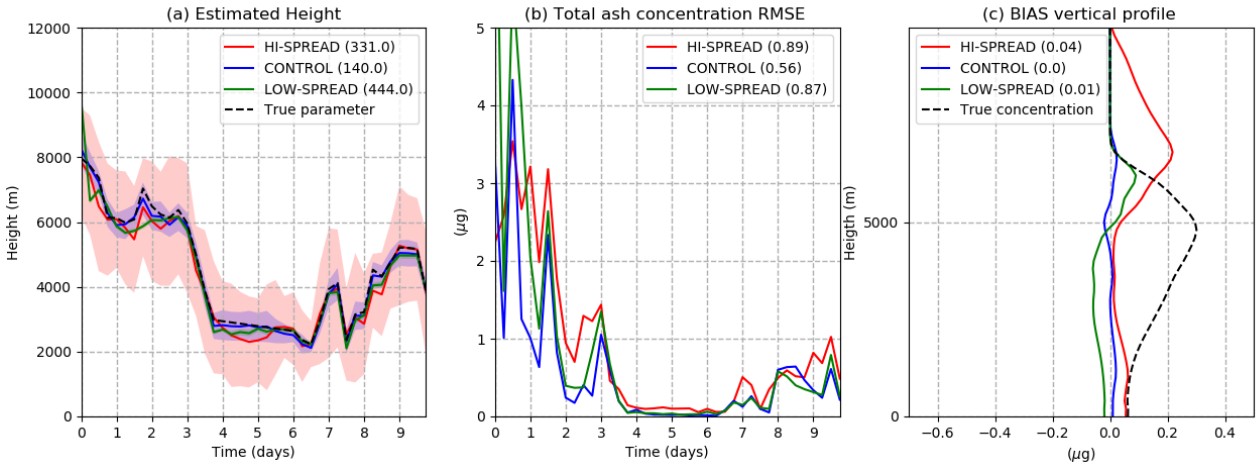

**Figure 9.** (a) Estimated $h$ as a function of time for the HI-SPREAD (red line), CONTROL (blue line) and LOW-SPREAD (green line). The shade surrounding the estimated values represents +/- one standard deviation and the black dashed line indicates the true parameter value. (b) Spatially averaged total ash concentration 6-hour forecast RMSE as a function of time. Line color code as in (a). (c) Temporally averaged 6-hour forecast bias as a function of height. Line color code as in (a).





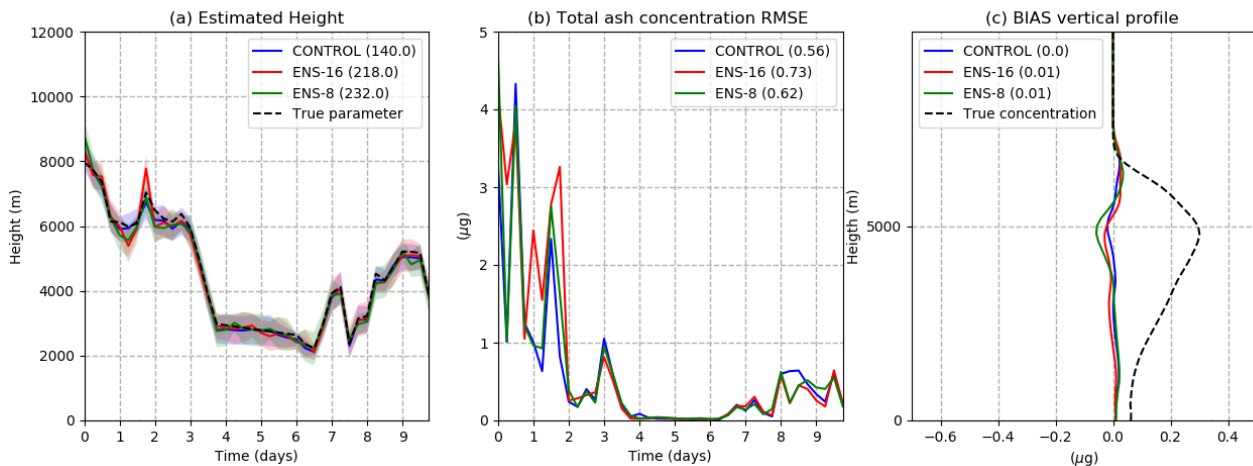

**Figure 10.** As in Figure 9 but for the experiments CONTROL, ENS-16 and ENS-8.



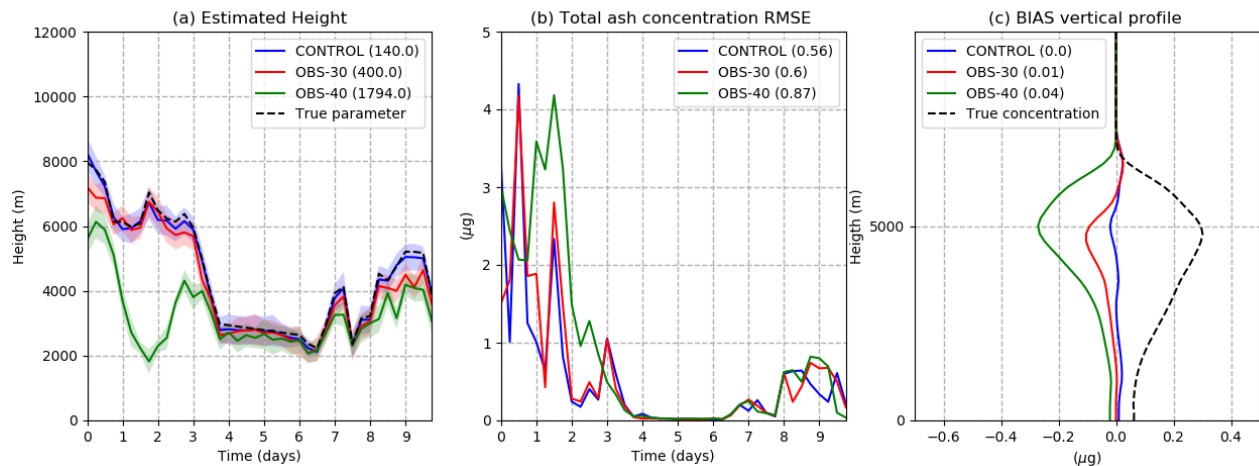

**Figure 11.** As in Figure 9 but for the experiments CONTROL, OBS-30 and OBS-40.



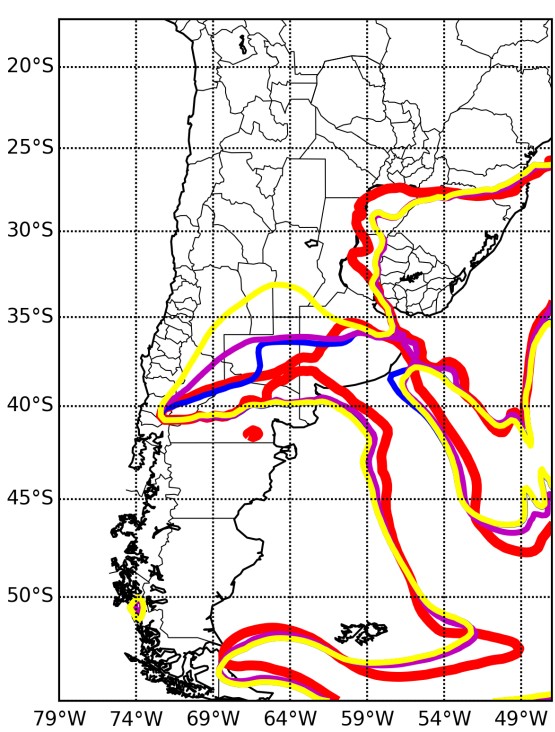

**Figure 12.** Ash mass loading 0.2 $gm^{-2}$ contours from: the nature run (red), ensemble mean optimized parameters run (blue) and the 12 (magenta) and 24 (yellow) hours forecasts verifying at the 8$^{th}$ of June at 00:00 UTC over a larger domain (see the text for details).

**Table 1.** Summary of the notation used in the paper. Nomenclature in ETKF method and their correspondence to the experiments discussed in this work. Where $n$ is the total number of grid points times the number of particle classes in the FALL3D model, $m$ is the number of observations at time $t$, $p$ is the number of parameters and $k$ is the number of ensemble members.

| Nomenclature | Dimension | Description | ETKF-FALL3D |
|---|---|---|---|
| $M_t$ | — | Non-linear model | FALL3D model |
| $\boldsymbol{y}_t^o$ | $m \times l$ | Observations | Satellite retrieval of ash mass loading |
| $\epsilon_t$ | $m \times l$ | Observational error | Ash mass loading estimation error |
| $\boldsymbol{x}_t^f$ | $n \times k$ | A priori or forecast ensemble | Ensemble forecast of 3D concentration |
| $\overline{\boldsymbol{x}}_t^f$ | $n \times l$ | Background mean | Mean of 3D concentration short term forecast |
| $\sigma_t^f$ | $p \times k$ | A priori or forecast set of parameters | A priori parameters ensemble used in the FALL3D forecast |
| $\boldsymbol{y}_t^f$ | $m \times k$ | Forecast into the observational space | FALL3D ash mass loading ensemble forecast |
| $\overline{\boldsymbol{y}}_t^f$ | $m \times l$ | Forecast mean | Ash mass loading ensemble forecast mean |
| $\boldsymbol{x}_t^a$ | $n \times k$ | A posteriori or analysis ensemble | Ensemble analysis of 3D concentration |
| $\overline{\boldsymbol{x}}_t^a$ | $n \times l$ | Analysis mean | Mean 3D concentration analysis |
| $\sigma_t^a$ | $p \times k$ | A posteriori or analysis set of parameters | Ensemble of optimized set of parameters |
| $\mathcal{H}_t$ | — | Observational operator | Transformation function from concentration to ash mass loading |
| $\mathbf{H}_t$ | $m \times n$ | Tangent linear observation operator | |
| $\mathbf{P}_t^f$ | $n \times n$ | Background error covariance matrix | 3D concentration forecast error covariance matrix |
| $\mathbf{P}_t^a$ | $n \times n$ | Analysis augmented state error covariance matrix | 3D concentration analysis error covariance matrix |
| $\mathbf{R}_t$ | $m \times m$ | Observational error covariance matrix | Ash mass loading error covariance matrix |
| $\boldsymbol{s}_t^f$ | $ns \times l$ | Augmented state vector | Concatenation of the state vector $\boldsymbol{x}_t$ and the estimated model parameters $\sigma_t^f$ |
| $\mathbf{S}_t^f$ | $ns \times ns$ | Ensemble forecast perturbation matrix | |



**Table 2.** Summary of the main parameters that distinguish the different experiments described in the text.

| Name | Ens. Size | $h$ ini. (m) | $A$ ini. | $h$ spread (m) | $A$ spread | Par. est. | Obs. Err. (%) |
|---|---|---|---|---|---|---|---|
| CONSTANT-UPPER | 32 | 11.000 | 7.0 | 500.0 | 2.0 | y | 15 |
| CONSTANT-LOWER | 32 | 3.000 | 2.0 | 500.0 | 2.0 | y | 15 |
| CONSTANT-NOEST | 32 | 3.000 | 2.0 | 500.0 | 2.0 | n | 15 |
| CONSTANT-TRUE | 32 | 8.500 | 5.5 | 500.0 | 2.0 | n | 15 |
| CONTROL | 32 | 11.000 | 7.0 | 500.0 | 2.0 | y | 15 |
| CONTROL-NOEST | 32 | 11.000 | 7.0 | 500.0 | 2.0 | n | 15 |
| HI-SPREAD | 32 | 11.000 | 7.0 | 2000.0 | 4.0 | y | 15 |
| LOW-SPREAD | 32 | 11.000 | 7.0 | 100.0 | 0.1 | y | 15 |
| ENS-16 | 16 | 11.000 | 7.0 | 500.0 | 2.0 | y | 15 |
| ENS-8 | 8 | 11.000 | 7.0 | 500.0 | 2.0 | y | 15 |
| OBS-30 | 32 | 11.000 | 7.0 | 500.0 | 2.0 | y | 30 |
| OBS-40 | 32 | 11.000 | 7.0 | 500.0 | 2.0 | y | 40 |





### Appendix A: ETKF formulation

A brief description of the ensemble transform Kalman filter equations are provided here. See Hunt et al. (2007) for a derivation of the equations as well as for a detailed discussion of the method. The ETKF approach solves the Kalman filter equations in the sub-space defined by the ensemble perturbations (i.e. the departures of individual members from the ensemble mean).

Under this framework, the update in the ensemble mean can be expressed as a linear combination of the forecast perturbations as follows:

$$\overline{s_t^a} = \overline{s_t^f} + \mathbf{S_t^f}\overline{w_t^a} \tag{A1}$$

where the $\overline{w_t^a}$ is a vector of weights of dimension $k$ computed as:

$$\overline{w_t^a} = \widetilde{\mathbf{P_t^a}}(\mathbf{Y_t^f})^T\mathbf{R}^{-1}(y_t - \overline{y_t^f}) \tag{A2}$$

Here $\mathbf{Y_t^f}$ is the ensemble perturbation matrix in observation space, whose i-th column is computed as $\mathbf{Y_t^{f(i)}} = \mathcal{H}(x_t^{f(i)}) - \mathcal{H}(x_t^f)$ and $\widetilde{\mathbf{P_t^a}}$ is the analysis covariance matrix in the sub-space spanned by the ensemble members and is computed as:

$$\widetilde{\mathbf{P_t^a}} = [(k-1)\mathbf{I} + (\mathbf{Y_t^f})^T\mathbf{R}^{-1}\mathbf{Y_t^f}]^{-1} \tag{A3}$$

$\mathbf{I}$ being the identity matrix of size $k \ x \ k$. The analysis ensemble perturbations are obtained as an optimal linear combination of the background ensemble perturbations:

$$\mathbf{S_t^a} = \mathbf{S_t^f}\mathbf{W_t^a} \tag{A4}$$

and the weight matrix $\mathbf{W_t^a}$ is computed as:

$$\mathbf{W_t^a} = [(k-1)\widetilde{\mathbf{P_t^a}}]^{1/2} \tag{A5}$$

Finally, the analysis ensemble is obtained as the sum of the analysis ensemble mean and the analysis perturbations:

$$s_t^{a(i)} = \overline{s_t^a} + \mathbf{S_t^{a(i)}} \tag{A6}$$

Note also that, in this implementation, the tangent linear observation operator $\mathbf{H}$ is not applied explicitly since $\mathbf{H_t}\mathbf{P_t^f}\mathbf{H_t^T}$ is approximated by $\mathbf{Y_t^f}(\mathbf{Y_t^f})^T$. Once the analysis ensemble for the augmented state is obtained, one can proceed to the next assimilation cycle.





*Author contributions.* All authors conceived the presented idea, designed the experiments and conducted the analysis of the results. AF provided guidance on the use of the FALL3D model and JR provided guidance on the implementation of the Local Ensemble Transform Kalman Filter. SO and JR developed the code and performed the computations. All the authors contributed to manuscript writing and approved the final manuscript.

5   *Acknowledgements.* S. Osores has been founded by a CONICET-CONAE doctoral fellowship. J. Ruiz has been funded by grants PICT2014-1000 and PICT2017-2233 of the Argentinian National Agency for Scientific Research Promotion and by grants UBACyT 2014 and 2018 from the University of Buenos Aires. Simulations were made with the high-performance computing clusters available at the Barcelona Super Computing Center, Spain and at CIMA/UBA-CONICET, Argentina. This work has been partially funded by the H2020 Center of Excellence for Exascale in Solid Earth (ChEESE) under the Grant Agreement 823844.





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
