# Peer review of "Volcanic ash forecast using ensemble-based data assimilation: the Ensemble Transform Kalman Filter coupled with FALL3D-7.2 model (ETKF-FALL3D, version 1.0)"

_Geoscientific Model Development, 2019_

## Referee Comment (RC1) · Anonymous Referee #1 · 14 Jul 2019

Review of "Volcanic ash forecast using ensemble-based data assimilation: the Ensemble Transform Kalman Filter coupled with FALL3D-7.2 model (ETKF-FALL3D, version 1.0)" by Soledad Osores et al.

General comment

The paper presents the application of an ensemble-based data assimilation algorithm, the Ensemble Transform Kalman Filter ETKF, to the FALL3D ash dispersion model. The model is tested against synthetic observations of volcanic ash concentration and

the authors show that the application of the kalman filter allows to better constraint column height, vertical mass distribution and atmospheric ash concentration.

The paper proposes an interesting and new application of data assimilation applied to volcanic eruptions. Indeed, despite widely used in atmospheric science and oceanography, ensemble-based data assimilation algorithms are almost unexplored within the volcanological community, with very few papers dealing with this topic. Overall the paper is well written and well structured. For these reasons I think it deserves publication after moderate revisions.

Some points could be improved and make clearer with the addition of more details and explanations. My specific comments are listed below.

Specific comments

P4L1: Specify what you mean for eruption column height. The maximum height or the neutral buoyancy.

P4L2: More details about the Suzuki vertical mass distribution should be given. A figure could be added to explain and visualize what A and lambda are. Moreover the assumptions and the limits of such distribution in term of loss of mass along the column should be added.

P7L1: Have you tested your model for wind field with spatial resolution different from 0.5? Have you noticed any change in the results? I think the sensitivity of the results to wind field resolution is an important aspect which should be tested and discussed.

P7L3 More details on how synthetic observations are generated should be given, maybe adding a figure showing both the observations and the errors.

P7L8 It is not clear why the authors consider observations with values between 0.2 and 10 gm-2 only. Please add explanations.

P8L7 Please give more details about covariance localization and why it is not necessary for this case.

P8L9 How many negative values do you observe? Is the number of negative values changing with model setting (ensemble number, spread, observation error, wind field resolution)?

P11L13 I think you cannot say that the A-Suzuky parameter is well estimated in this case, please explain.

P14L19 "The experiments focused on two FALL3D model parameters, one that defines the vertical emission profile and the eruptive column height (and related emitted mass)." The sentence should be rephrased.

P31L2 Is the filter a Local one? I thought it was a global filter.

Figures

In the main text, figures are indicated both with Fig. and Figure. I think only one notation should be used.

Figure 8: Could you explain the measurement unit of the concentration? Are you missing m2?

Figure 12: I found Figure 12 not easy to read. In particular the blue contour is not easy to follow. Maybe the figure could be split into 4 panels showing the ash mass loading for the 4 cases.

---

## Referee Comment (RC2) · Fei Lu (Referee) · 30 Aug 2019

The paper provides a nice application of data assimilation method to volcanic ash forecast, using the Ensemble Transform Kalman Filter with the FALL3D model. In particular, the paper investigates the possibility of estimating the parameters in the model, so as to reduce uncertainties related to eruption source parameters. The author reported that the joint estimation of concentration and source parameters lead to better analysis and forecast of the 3D ash concentrations.

Parameter estimation is an important part of model development. The paper investigated different scenarios: static parameters, time-dependent parameters, and sensitivity analysis. Such a systematic study of joint parameter state estimation will be helpful for further development of volcanic ash cloud modeling. Therefore, I recommend its publication at GMD. But a few important details seem missing:

- What is the model for the parameter flow in the ETKF? In page 4 line 30, the authors mentioned that "a persistence model is assumed for the model parameters (i..e $\theta_t^f = \theta_{t-1}^a$)". This would mean that the model for the parameter flow is $\theta_t = \theta_{t-1}$, and the ensemble of the parameters will only shrink, which does not agree with the plots in Figure 4, where the ensemble oscillates as if $\theta_t = c\theta_{t-1} + W_t$.

- How is the ensemble of the parameter generated at the initial time? How is the physical constraint (page 8 line 15) ensured in the spread of the ensemble at all times? The physically meaningful range (page 8 line 15) is 0-20 km and 0-15, but the spreads in tests are 500m and 0.5 (page 8 line 24). Is there any specific reason for such a relatively small spread?

- Is it possible to describe how does the FALL3D model depend on the parameters? Are there some parameters lead to instability or unphysical state values?

---

## Author Comment (AC1) · 27 Sep 2019

We would like to thank the reviewer for the comments provided which helped to significantly improve the clarity of the manuscript. Below you will find the answers to the specific comments. Reviewers comments are indicated in bold letters. Changes in the manuscript are indicated in blue.

1) **P4L1: Specify what you mean for eruption column height. The maximum height or the neutral buoyancy.**

We agree with the reviewer, this point was not clear in the original version of the manuscript. Following this comment we include a clarification in Page 4 Line 1:

"which depends on the fourth power of the top height of the eruptive column and does not account for wind effects"

2) **P4L2: More details about the Suzuki vertical mass distribution should be given. A figure could be added to explain and visualize what A and lambda are. Moreover the assumptions and the limits of such distribution in term of loss of mass along the column should be added.**

We agree with both reviewers, more information about the sensitivity of FALL3D to the selected parameters is required to better understand the results. Following this comment that is also shared by Reviewer 2 a brief description of the Suzuki parametrization of the source term has been added and a Figure showing the sensitivity of the source to this parameters is included in the text. Also a reference is included where the sensitivity of the model solution to this parameters has been explored in more detail. These changes are included in Page 4, between Lines 2 and 17 and a new Figure (Figure 1) has been prepared which shows the sensitivity of the vertical ash emission profile to parameters h and A (which are the ones estimated in this work):

"For simplicity and without loss of generality, we will assume here a MER given by the Mastin et al., (2009) scheme, which depends on the fourth power of the top height of the eruptive column and does not account for wind effects, and a Suzuki vertical mass distribution (Pfeiffer et al., 2005) that is an empirical vertical ash mass eruption rate distribution that assumes no interactions with the surrounding atmosphere (e.g. effects of wind shear or stratification upon the eruptive column), also it is assumed that the shape of the vertical flow rate is the same for all particle sizes and is given by:

$$S(z) = (1 - \tfrac{1}{h} exp[A\tfrac{z}{h} - 1])^{\lambda} \qquad\qquad\qquad (1)$$

where S(z) is the mass eruption rate distribution function, z is the altitude above the vent, h is the top height of the eruptive column, A and $\lambda$ are two dimensionless parameters. Figure 1 shows the sensitivity of the vertical emission profile to different values of h and A. It is important to recall that h not only controls the maximum height of the eruptive column, but also the total mass emitted (Fig. 1 a). Parameter A do not modify the total amount of mass being emitted but significantly affects the level at which the maximum emission takes place (Fig. 1 b) which can significantly affect the posterior evolution of the ash plume particularly if cases in which there is strong vertical wind shear. The parameter $\lambda$ is a measure of how concentrated is the emission around the maximum (not shown). A previous sensitivity test (Osores,2018) has shown that the two FALL3D model parameters that affect most the model results are the eruption column height h and the parameter A in the Suzuki distribution. For this reason, these two parameters will be used in the following sections to define the ETKF-FALL3D system experiments. The sensitivity of the FALL3D model to these parameters in terms of the deposit, has been documented by (e.g., Scollo et al., 2008) ."

[Figure]

Figure 1. Vertical mass distribution for different (a) eruptive column top heights and (b) A-Suzuki parameters

**3) P7L1: Have you tested your model for wind field with spatial resolution different from 0.5? Have you noticed any change in the results? I think the sensitivity of the results to wind field resolution is an important aspect which should be tested and discussed.**

We agree with the reviewer, this is an interesting point. A discussion of parameter estimation in the presence of model or boundary conditions errors is important for parameter estimation in real-life applications in which there are several sources of model imperfection. Model resolution or in this case the quality of the meteorological forcing (associated of the horizontal resolution of the meteorological model) is one of the sources of imperfection in offline dispersion models.

While we believe that a detailed description of the impact of model error upon the optimization if beyond the scope of this paper which deals with a description of the technique, a first assess under idealized conditions and also a sensitivity study to its parameters, we perform an experiment that shows that the technique is robust to errors in the meteorological field.

We performed an experiment in which the meteorological forcing is provided by the NCEP-NCAR Reanalysis V2 with 2.5 degree horizontal resolution (much lower than the one used to generate the "true run"). We repeated the estimation of the parameters using this meteorological forcing under the constant parameter scenario (experiment CONSTANT-RE). Figures 1 and 2 summarizes the results.

[Figure]

*Figure 1: Optimized parameters as a function of time in the CONSTANT-UPPER (blue line), CONSTANT-LOWER (red line), CONSTANT-TRUE (black line), CONSTANT-NOEST (green line) and CONSTANT-RE (magenta line) experiments. The shade surrounding the CONSTANT-UPPER and CONSTANT- LOWER estimated values represents +/- one standard deviation. (a) h parameter and (b) A − Suzuki parameter.*

[Figure]

*Figure 2: Spatially averaged forecasted total ash concentration forecast RMSE for the CONSTANT-UPPER (blue line), CONSTANT-LOWER (red line), CONSTANT-TRUE (black line), CONSTANT-NOEST (green line) and CONSTANT-RE (magenta line) experiments. All values are in $10^{-3}\,gm^{-3}$ .*

Figures 1 and 2 show that the estimation provided in the CONSTANT-RE experiment is worse than the one provided in the CONSTANT-UPPER or CONSTANT-LOWER experiments where the forcing is perfect (as expected). However the estimation provided by CONSTANT-RE experiment is stable and provides much better results than the case in which the source parameters are not estimated at all.

Although these results are not included in the revised version of the manuscript, a comment is included in the Summary and Conclusions section (see Page 15, Lines 21-26):

"Experiments discussed in this work assumes a perfect model and a perfect meteorological forcing. In real-life applications, imperfections in the model and the forcing has a significant impact on the quality of ash dispersion forecasts. Previous works have shown that parameter estimation can be successfully performed in the presence of multiple sources of model error (e.g. Ruiz and Pulido,

2015). Preliminary experiments introducing errors in the meteorological forcing suggest that the current system provides a robust estimation of the source parameters in the presence of wind uncertainty. However, this aspect should be further analyzed in future studies."

**4) P7L3 More details on how synthetic observations are generated should be given, maybe adding a figure showing both the observations and the errors.**

We splitted Figure 3 in two Figures, now Fig. 4 and 5, including in Fig. 5 the Nature Run and the observation simulation for the constant parameter and variable parameters runs to show the difference between them. A brief description of the impact of the error against the nature run is included in Page 8 Lines 2-16.

**3.1.1 Constant emission profile**

This nature run simulation considers a source term that remains constant during all the simulated period, with an eruption column height of 8.5 km above the vent and a A-Suzuki parameter of 5.5 (Fig. 4). Figure 5 (a) and (c) shows the ash mass loading from the nature run and the observation simulation on 7$^{th}$ June at 12:00 UTC for illustrative purposes. The addition of the observational error to the nature run does not significantly affect the spatial distribution nor the location and intensity of the maximum concentration. The number of available observations (which depends on the thresholds described in the previous section) is time-dependent (ranging from 27 to 52 grid point observations) and, in this particular case, is primarily affected by the atmospheric circulation which produces variations in the 3D ash concentration within the model domain.

**3.1.2 Variable emission profile**

In this experiment, h and A-Suzuki are time dependent (Fig. 4). In order to represent a realistic variability of the source parameters, the h evolution corresponds to the estimated heights during the 2011 Puyehue-Cordón Caulle eruption (Osores et al., 2014). Since the A-Suzuki parameter can not be directly estimated, the evolution of this parameter is simulated using an auto-regressive model (Fig. 4).

In Fig. 5 (b) and (d), the 7$^{th}$ June at 12:00 UTC ash mass loading fields from the nature run and the observation simulation are shown. As has been shown for the constant parameters case, the observational error does not affect significantly the spatial distribution of the plume. In this experiment, the number of observations that are assimilated depends on the emission profile as well as the wind field, and it can range from 15 (on 11$^{th}$ June 06:00 UTC) to 86 (on 11$^{th}$ June at 18:00 UTC).

[Figure]

Figure 4. Nature run parameters time series for the constant (solid lines) and variable emission profiles (dashed lines) for h (black lines) and A-Suzuki (red lines).

[Figure]

Figure 5. Ash mass loading on 7th June 12:00 UTC for the (a) constant parameters nature run and (c) constant parameters run with observational error and the (b) time dependent parameters nature run and (d) time dependent parameters run with observational error. Ash mass loading values outside the 0.2-10.0 g m-2 interval are in grey.

**5) P7L8 It is not clear why the authors consider observations with values between 0.2 and 10 gm-2 only. Please add explanations.**

We agree with the reviewer and more information has been provided to clarify this point. The text has been modified at Page 7, Lines 22-24:

"To represent some of the limitations of current satellite-based ash mass load retrievals, the simulated observations are available only where the true load values are between 0.2 g m$^{-2}$ and 10 g m$^{-2}$. The lower bound approximately corresponds to the minimum mass load that can be retrieved by the state-of-the-art algorithms. Retrievals usually can not estimate mass loads over the upper bound because the optical thickness of the corresponding ash plume is too high (e.g., Wen and Rose, 1994; Prata and Prata 2012; Pavolonis et al., 2013). The observational error is assumed to have a random Gaussian distribution, with a standard deviation of 0.15 of the ash mass load."

**6) P8L7 Please give more details about covariance localization and why it is not necessary for this case.**

We agree with the reviewer in that this issue deserves further explanation. The text in page 8 Lines 31 to Page 9 Line 2 has been modified as follows:

"Covariance localization is usually required to reduce the impact of spurious correlation that results from the use of small ensemble sizes. The estimation of small correlations (e.g. between locations that are far apart from each other) is usually strongly affected by sampling noise, this is why estimated covariances are usually forced to decay with distance. Since the domain used in the data assimilation experiments is small, the impact of spurious correlations between distant grid points is less significant. For this reason, no covariance localization is used in the estimation of the state variables or the parameters. However, is important to keep in mind that if the system is extended to larger domains using covariance localization will highly improve its performance."

**7) P8L9 How many negative values do you observe? Is the number of negative values changing with model setting (ensemble number, spread, observation error, wind field resolution)?**

Following the reviewer's comment we provide more information about this particular point. We evaluate the proportion of ensemble members in which the ash concentration or the parameter values needs to be corrected because they fall outside the physically meaningful range. We evaluate this rate in the time dependent parameter scenario (to extract the sensitivity to the parameter value) and for two different parameter ensemble spreads.
Based on the obtained results the text has been modified as follows (See Page 9 and Line 11-15):

"… The physically meaningful range for model parameters is set to 0-20 km and 0-15 for h and A-Suzuki respectively..
The number of grid points and ensemble members with estimated concentrations below -1.0 e$^{-4}$ g m$^{-3}$ is usually below 15% of the grid points and ensemble members at which concentration has been updated. This proportion decreased with increasing ash concentration as well as with ensemble spread. Estimated parameters for individual ensemble members fall outside the physical meaningful range less than 10% of the times also depending on how close to the boundaries are the true parameters and how large is the parameter ensemble spread."

**8) P11L13 I think you cannot say that the A-Suzuky parameter is well estimated in this case, please explain.**

We agree with the reviewer and following this comment the text has been modified as follows (see Page 12 Line 15-16)

"For the A-Suzuki parameter, the time evolution is not reproduced so accurately. There are also two sudden jumps in the estimation of A-Suzuki, indicating a less well constrained parameter value."

**9) P14L19 "The experiments focused on two FALL3D model parameters, one that defines the vertical emission profile and the eruptive column height (and related emitted mass)." The sentence should be rephrased.**

We have rephrased the sentence to (Page 15 Line 24-26):

 "The experiments focused on two FALL3D model parameters, one that defines the vertical emission profile and, one defining the top height of the eruptive column. The last one is also related to emitted mass."

**10) P31L2 Is the filter a Local one? I thought it was a global filter.**

Thank you very much for detecting this error. We have modified the text accordingly in Page 35, Line 2.

**Figures**

**In the main text, figures are indicated both with Fig. and Figure. I think only one notation should be used.**

Thank you very much for detecting this issue. The use of Figure and Fig. through the text was corrected following  the suggested format at the journal's web site:
https://www.geoscientific-model-development.net/for_authors/manuscript_preparation.html
Regarding the use of Figure and Fig. it says:
"The abbreviation "Fig." should be used when it appears in running text and should be followed by a number unless it comes at the beginning of a sentence, e.g.: "The results are depicted in Fig. 5. Figure 9 reveals that..."

**Figure 8: Could you explain the measurement unit of the concentration? Are you missing m2?**

Thank you for pointing this error. We have corrected the units in Figure 8 (Figure 10 in the new version of the manuscript) to g m$^{-3}$.

**Figure 12: I found Figure 12 not easy to read. In particular the blue contour is not easy to follow. Maybe the figure could be split into 4 panels showing the ash mass loading for the 4 cases.**

We improve the previous Figure 12 (now Figure 14) showing a comparison of ash mass loading above 0.2 g m$^{-2}$ between the simulations using the estimated parameters, the 12 and 24 hours forecast against the nature run. The text in Page 15 line 1-9, was adapted as follows:

"To see if the estimated parameters can be used to reconstruct the ash cloud far from the source, the estimated parameters are used to produce a simulation of the ash cloud over a larger domain. At each time the source parameter values in this simulation are taken from the CONTROL run parameter ensemble mean. This simulation will be referred as CONTROL-LD. Figure 14 (a) shows the results of comparing the ash mass loading above 0.2 g m$^{-2}$ from the experiment forced with the estimated parameters against the nature run. The comparison of these categorical variables shows that hits (i.e. grid points in which mass loadings are over the selected threshold for both, the simulation and the nature run) prevail, with a lower number of false alarms and misses (i.e. grid points in which the simulation is over the threshold and the nature is not or vice-versa respectively). We note that both ash clouds are very close to each other even far from the source, indicating that the estimated parameters are sufficient for the reconstruction of the ash plume in this ideal case.

To see if the CONTROL-LD experiment can be used to initialize short range ash concentration forecast over the larger domain a forecast is initialized using the CONTROL-LD ash concentrations as initial conditions and the CONTROL parameter ensemble mean as source parameters. Note that in this case, parameters remain constant during the forecast. Figures 14 (a) and (b) show the 12 and 24 hours forecast lead times initialized on 7$^{th}$ of June at 12:00 UTC and 00:00 UTC respectively. There is a good agreement between forecasts and the nature run. For larger lead times there are more false alarms and misses as expected. This suggests that initializing a forecast from a long run forced with the optimized parameters can be a cost-effective strategy to generate short lead ash concentration forecast over a relatively large domain. "

[Figure]

Figure 14. Ash mass loading above 0.2 g m$^{-2}$ comparison between the ensemble mean optimized parameters run, the 12 and 24 hours forecasts against the nature run respectively over a larger domain verifying at the 8$^{th}$ of June at 00:00 UTC (see the text for details)

---

## Author Comment (AC2) · 27 Sep 2019

We would like to thank the reviewer for the comments provided which helped to significantly improve the clarity of the manuscript. Below you will find the answers to the specific comments.

Reviewers comments are indicated in bold letters. Changes in the manuscript are indicated in blue.

1) **What is the model for the parameter flow in the ETKF? In page 4 line 30, the authors mentioned that "a persistence model is assumed for the model parameters (i..e θ f t = θ a t-1 )". This would mean that the model for the parameter flow is θt = θt-1, and the ensemble of the parameters will only shrink, which does not agree with the plots in Figure 4, where the ensemble oscillates as if θt = cθt-1 + Wt .**

We agree with the reviewer that this point was not clear in the text. Parameters are assumed to remain constant during model integration. However after assimilation we performed multiplicative inflation so that the analysis parameter ensemble spread equals the first-guess parameter ensemble spread. In this way we avoid the collapse of the ensemble of parameters and can estimate their time evolution. We add included this in Page 8, Lines 26-28.

"The relaxation to prior spread inflation approach (RTPS, Whitaker and Hamill (2012)) with a parameter of $\alpha$ =0.5 is applied to the state variables to reduce the impact of sampling error. For the parameters, the ensemble spread is inflated back to its original value after assimilating the observations (similar to the conditional inflation approach of Aksoy et al. (2006). This is equivalent to assume that the parameter uncertainty is time-independent, thus preventing the parameter ensemble spread from collapsing."

2) **How is the ensemble of the parameter generated at the initial time?**

We thank the reviewer for pointing this out, we add some clarification to the text, in Page 8, Line 24 and Page 9, Line 4-5.

"The ensemble at the first assimilation cycle is initialized using zero ash concentrations for all members and a set of parameters that are sampled randomly from a Gaussian distribution whose mean and variance for each experiment are detailed below."

...

"All presented data assimilation and parameter estimation experiments are summarized in Table 2 including the statistical properties of the initial parameter ensemble. Finally, a set of simulation experiments are carried out using a larger domain to evaluate the impact of the optimized parameters upon the simulation of the ash cloud farther from the vent."

**3) How is the physical constraint (page 8 line 15) ensured in the spread of the ensemble at all times?**

We thank the reviewer for pointing this out. The explanation given in the text was not clear. The check of the parameter values is performed individually for each ensemble member. An improved description of the algorithm is provided in Page 9, Line 6-10.

"In the case of eruption source parameters, nonphysical values are checked individually for each ensemble member and are replaced with a random realization from a Gaussian distribution with the same mean and standard deviation as the analysis ensemble. If the randomly generated value is outside the physically meaningful range for the parameter, the process is repeated until the randomly generated value is within the physically meaningful range. The physically meaningful range for model parameters is set to 0-20 km and 0-15 for h and A-Suzuki respectively."

**4) The physically meaningful range (page 8 line 15) is 0-20 km and 0-15, but the spreads in tests are 500m and 0.5 (page 8 line 24). Is there any specific reason for such a relatively small spread?**

We thank the reviewer for raising this point. Following the reviewer's comment we add a short discussion about this in Page 11, Lines 16-20.

"Figure 7 (b) shows the spatially averaged ash concentration ensemble spread. One way to assess if the current parameter ensemble spread is well tuned is to compare the ash concentration forecast error and spread. If these are similar then we can assume that our uncertainty is well represented in the ensemble. In this case, the uncertainty in the ash concentration is mainly associated with the uncertainty in the source parameters. As observed, the spread values are close to the RMSE values in Figure 7 (a), which indicates that after convergence of the parameters, the ensemble spread is closely representing the magnitude of the errors."

**5) Is it possible to describe how does the FALL3D model depend on the parameters?**

Following this comment that is also shared by Reviewer 1, we added a line citing an article that makes a deep discussion on the sensitivity of the model to different parameters. Also we extended the explanation of Suzuki source term and how it affects the mass distribution in the vertical profile. These changes are included in Page 4, between Lines 2 and 17 and a new Figure (Figure 1) has been prepared which shows the sensitivity of the vertical ash emission profile to parameters h and A (which are the ones estimated in this work):

"For simplicity and without loss of generality, we will assume here a MER given by the Mastin et al., (2009) scheme, which depends on the fourth power of the top height of the eruptive column and does not account for wind effects, and a Suzuki vertical mass distribution (Pfeiffer et al., 2005) that is an empirical vertical ash mass eruption rate distribution that assumes no interactions with the surrounding atmosphere (e.g. effects of wind shear or stratification upon the eruptive column), also it is assumed that the shape of the vertical flow rate is the same for all particle sizes and is given by:

$$S(z) = (1 - \tfrac{1}{h}exp[A\tfrac{z}{h} - 1])^{\lambda} \qquad\qquad (1)$$

where S(z) is the mass eruption rate distribution function, z is the altitude above the vent, h is the top height of the eruptive column, A and $\lambda$ are two dimensionless parameters. Figure 1 shows the sensitivity of the vertical emission profile to different values of h and A. It is important to recall that h not only controls the maximum height of the eruptive column, but also the total mass emitted (Fig. 1 a). Parameter A do not modify the total amount of mass being emitted but significantly affects the level at which the maximum emission takes place (Fig. 1 b) which can significantly affect the posterior evolution of the ash plume particularly if cases in which there is strong vertical wind shear. The parameter $\lambda$ is a measure of how concentrated is the emission around the maximum (not shown). A previous sensitivity test (Osores, 2018) has shown that the two FALL3D model parameters that affect most the model results are the eruption column height h and the parameter A in the Suzuki distribution. For this reason, these two parameters will be used in the following sections to define the ETKF-FALL3D system experiments. The sensitivity of the FALL3D model to these parameters in terms of the deposit, has been documented by (e.g., Scollo et al., 2008) ."

[Figure]

Figure 1. Vertical mass distribution for different (a) eruptive column top heights and (b) A-Suzuki parameters

**6) Are there some parameters lead to instability or unphysical state values?**

This is a good point. Parameters optimized in this work are related to the eruptive source. Because of this there are no instabilities associated to the parameters. However, unphysical state values can result for example from negative parameter values (e.g. negative column height or A-Suzuki parameter).